# OMNIEDUBENCH: A COMPREHENSIVE CHINESE BENCHMARK FOR EVALUATING LARGE LANGUAGE MODELS IN EDUCATION

## ABSTRACT

With the rapid development of large language models (LLMs), various LLM-based works have been widely applied in educational fields. However, most existing LLMs and their benchmarks focus primarily on *the knowledge dimension, largely neglecting the evaluation of cultivation capabilities* that are essential for real-world educational scenarios. Additionally, current benchmarks are often *limited to a single subject or question type, lacking sufficient diversity*. This issue is particularly prominent within the Chinese context. To address this gap, we introduce **OmniEduBench, a comprehensive Chinese educational benchmark**. OmniEduBench consists of 24.602K high-quality question-answer pairs. The data is meticulously divided into two core dimensions: **the knowledge dimension and the cultivation dimension**, which contain 18.121K and 6.481K entries, respectively. Each dimension is further subdivided into 6 fine-grained categories, covering a total of 61 different subjects (41 in the knowledge and 20 in the cultivation). Furthermore, the dataset features a rich variety of question formats, including 11 common exam question types, providing a solid foundation for comprehensively evaluating LLMs' capabilities in education. Extensive experiments on 11 mainstream open-source and closed-source LLMs reveal a clear performance gap. In the knowledge dimension, only Gemini-2.5 Pro surpassed 60% accuracy, while in the cultivation dimension, the best-performing model, QWQ, still trailed human intelligence by nearly 30%. These results highlight the substantial room for improvement and underscore the challenges of applying LLMs in education.

## 1 INTRODUCTION

With the rapid emergence of large language models (LLMs), evaluation benchmarks have become increasingly critical, shifting the focus of assessment toward broader and complex skills. To address the demands of this complex paradigm, a variety of benchmarks have been proposed to evaluate the diverse capabilities of LLMs. These benchmarks cover a wide spectrum of areas, including knowledge and language understanding (*e.g.*, MMLU (Hendrycks et al., 2021a), ARC (Clark et al., 2018)), reasoning (*e.g.*, GSM8K (Cobbe et al., 2021), AIME (Patel et al., 2024)), multi-turn open-ended dialogue (*e.g.*, MT-bench (Bai et al., 2024)), and coding (*e.g.*, MBPP (Austin et al., 2021)). Serving as indispensable tools for advancing LLM development, these benchmarks have been widely adopted in recent influential works (Hurst et al., 2024; Liu et al., 2024a; Team, 2025b; Comanici et al., 2025; Taylor et al., 2022; Touvron et al., 2023; OpenAI, 2023; Hoffmann et al., 2022).

In recent years, a series of powerful Chinese LLMs emerged, such as the Qwen (Yang et al., 2024; 2025), DeepSeek (Liu et al., 2024a; Guo et al., 2025; Liu et al., 2024b), achieving performance levels comparable to overseas LLMs. With the growing application of LLMs in education, researchers have also begun to propose Chinese education benchmarks, which can be broadly categorized into two types: (1) datasets translated from other languages and (2) datasets natively constructed from Chinese education corpora. Specifically, (1) Datasets translated from other languages refer to benchmarks constructed by directly translating existing benchmarks from other languages into Chinese. A representative work is CLUE (Xu et al., 2020a), which was translated from the English GLUE (Wang et al., 2019). However, a simple translation approach is insufficient for a rigorous evaluation of LLMs in Chinese. These datasets often fail to reflect the unique linguistic and cultural challenges

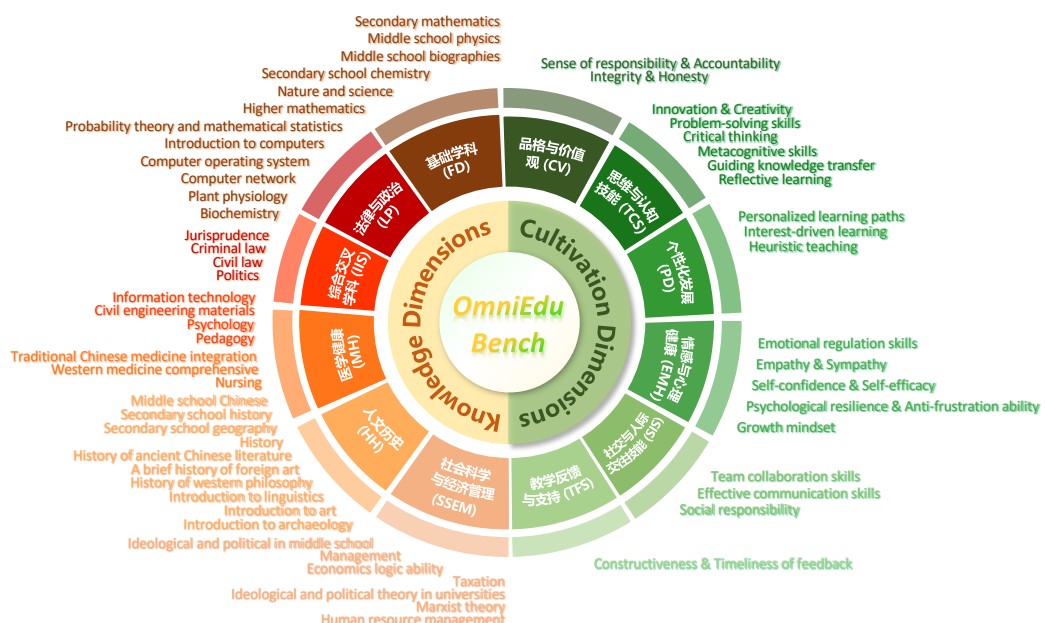

Figure 1: Overview of OmniEduBench. The benchmark comprises two dimensions: 41 subjects across six categories in the knowledge, and 20 subjects across six categories in the cultivation.

of the Chinese education and inherently carry biases from their original environment, thus limiting their ability to assess LLMs' understanding of local education knowledge and teacher-student needs.

(2) Datasets natively constructed from Chinese educational corpora refer to benchmarks directly collected from Chinese educational text resources, such as C-Eval (Huang et al., 2023), Edubench (Xu et al., 2025), Scieval (Sun et al., 2024), AGIEval (Zhong et al., 2023), and SuperCLUE (Xu et al., 2023). However, most existing education benchmarks are often limited to a single subject or question type, lacking sufficient diversity. Additionally, these datasets typically focus on the knowledge dimension, overlooking the unique cultivation aspects that are essential in real-world education.

We present OmniEduBench, a comprehensive Chinese education benchmark designed to thoroughly evaluate LLMs in terms of both knowledge understanding and skill cultivation in educational scenarios. OmniEduBench encompasses knowledge and cultivation dimension and comprises a total of 24.602K high-quality question–answer pairs, covering 11 common exam question types (*e.g.*, multiple choice (单选题), multiple answer (多选题), fill-in-the-blank (填空题), short answer (简答题), composite questions (复合题), term explanation (名词解释), True/False (判断题), calculation (计算题), logical reasoning (逻辑推理题), case analysis (案例分析题), and essay (论述题)), as illustrated in Figure 1. The knowledge dimension includes 18.121K question–answer pairs spanning 41 subject areas, from humanities to science and engineering, and covering five difficulty levels: elementary school, middle school, high school, college, and professional examinations. The cultivation dimension comprises 6.481K question–answer pairs across 20 teaching-related comments, including guided teaching, student emotional support, and moral education (see Table 1 for details), aiming to comprehensively assess the diverse competencies required in real-world educational settings. Extensive experiments demonstrate that our proposed OmniEduBench presents a highly challenging and significant benchmark for Chinese educational evaluation. Additionally, we introduce OmniEduBench HARD, a high-difficulty subset of OmniEduBench, specifically targeting particularly demanding subjects such as advanced mathematics and competitions that require sophisticated reasoning skills. Even the state-of-the-art LLMs achieve less than 50% accuracy on this subset, highlighting the rigor and necessity of our proposed OmniEduBench education benchmark.

## 2 RELATED WORK

In this section, we present a comprehensive survey of large language models (LLMs) and benchmarks related to our constructed OmniEduBench, encompassing both English and Chinese datasets.

## 2.1 LARGE LANGUAGE MODELS

Recently large language models have advanced at an unprecedented pace. Leveraging increasingly sophisticated architectures and ever-larger pretraining corpora, they have continuously pushed the boundaries of performance in language understanding, reasoning, and generation tasks. Researchers have explored various approaches to enhance LLMs' capabilities. For example, Chain-of-Thought prompting (Wei et al., 2022; Team, 2025c;b; Guo et al., 2025; Liu et al., 2024a) has been shown to be highly effective in guiding models to perform step-by-step reasoning for complex problem-solving. In addition, instruction tuning (Dong et al., 2025; Hu et al., 2025; Yang et al., 2024; 2025) and reinforcement learning from human feedback (RLHF) (Ouyang et al., 2022; Schulman et al., 2017) have been widely adopted to align model outputs with human intentions and preferences, enabling LLMs to generate responses that are more natural and reliable in open-ended dialogue and creative tasks. Despite these remarkable advances, however, the question of how to comprehensively and effectively evaluate the true capabilities of LLMs remains a critical and open challenge.

## 2.2 ENGLISH EDUCATION BENCHMARKS

Researchers have proposed a variety of benchmarks to evaluate the capabilities of LLMs, which can be broadly categorized into three types: (1) task-specific evaluations, such as reading comprehension (SQuAD (Rajpurkar et al., 2016)), machine translation (Bojar et al., 2014)), and summarization (Hermann et al., 2015); (2) general knowledge and advanced ability evaluations, for example, the Massive Multitask Language Understanding (MMLU) benchmark (Hendrycks et al., 2021a), which collects questions from real-world exams and textbooks to provide a diverse, multi-domain test that effectively probes the breadth and depth of model knowledge. Similarly, the BIG-bench benchmark (Srivastava et al., 2022) comprises 204 diverse tasks; and (3) specialized ability evaluations. In mathematical reasoning, benchmarks such as GSM8K (Cobbe et al., 2021) and MATH (Hendrycks et al., 2021b) assess models' ability to solve complex multi-step problems. In code generation, HumanEval (Chen et al., 2021) and MBPP (Austin et al., 2021) have become standard benchmarks for measuring programming proficiency. Additionally, datasets such as MT-bench (Zheng et al., 2023) have been introduced to evaluate performance in multi-turn, open-ended dialogues. Despite the significant contributions of these datasets to advancing LLMs evaluation, most of them remain heavily focused on English, with limited coverage of Chinese scenarios.

## 2.3 CHINESE EDUCATION BENCHMARKS

A series of comprehensive Chinese benchmarks have been proposed. For example, CLUE (Xu et al., 2020b), as an early work, integrates multiple natural language understanding tasks and has become an important reference for evaluating LLMs. Subsequently, benchmarks such as CMMLU (Li et al., 2023) and C-Eval (Huang et al., 2023) collect multi-disciplinary, multi-task questions from Chinese university exams, professional qualification tests, and textbooks, effectively assessing models' general knowledge and their understanding. Beyond general capability evaluation, researchers have also developed Chinese benchmarks targeting specific advanced skills. For example, in mathematical reasoning, CMATH (Wei et al., 2023) tests models' abilities to solve complex mathematical problems. Meanwhile, EduBench (Xu et al., 2025) constructs synthetic corpora for the education, but its question types are relatively limited, making it difficult to fully capture models' Chinese potential. To address this critical gap, we propose OmniEduBench — a comprehensive Chinese education benchmark that uniquely combines knowledge and nurturing dimensions, providing a novel, holistic framework for systematically evaluating LLMs' potential as educational assistants.

## 3 OMNIEDUBENCH

Our proposed OmniEduBench education benchmark is designed as a natively Chinese education evaluation benchmark that captures the unique linguistic and cultural knowledge of Chinese education, encompasses diverse question types, and assesses LLMs not only on their knowledge capabilities but also on the distinctive cultivation competencies required in real-world educational scenarios.

Figure 2: Overview of the construction process, including collection, cleaning, filtering, verification.

## 3.1 TASK DEFINITION

**Knowledge dimension** focuses on evaluating the model's mastery of subject-specific knowledge. Tasks in this dimension include 11 common exam question types (*e.g.*, multiple choice, multiple answer, fill-in-the-blank, short answer, composite questions, term explanation, True/False, calculation, logical reasoning, case analysis, and essay). These 11 question types span a wide range of disciplines, from humanities and history to science, engineering, and professional fields. The primary goal is to assess the LLM's problem-solving capabilities within the context of real-world education.

**Cultivation dimension** assesses LLMs on their ability to support holistic educational objectives beyond mere knowledge acquisition. This includes guiding students' thinking processes, fostering moral and value development, enhancing emotional understanding, and promoting critical reasoning skills. Tasks in this dimension are designed to reflect realistic learning scenarios, where models must provide pedagogically sound feedback that aligns with students' cognitive and emotional needs.

## 3.2 BENCHMARK CONSTRUCTION

In this section, we provide a detailed overview of the construction process for the proposed OmniEduBench education evaluation benchmark, as illustrated in Figure 2. The process consists of four key stages: dataset collection, dataset cleaning, dual-machine filtering, and expert verification.

**Dataset collection.** OmniEduBench is designed to encompass a wide range of diverse scenarios to enable comprehensive evaluation. To achieve this, we employ three distinct data collection methods, carefully balancing diversity and efficiency in the construction of the OmniEduBench benchmark.

*Manual collection of publicly available data.* Existing benchmarks often lack sufficient diversity in question types and knowledge coverage, making them inadequate for our 41 subjects in knowledge dimensions. To address this gap, we manually collected additional data from publicly available online resources (*e.g.*, XuekeNet, ZujuanNet, ShijuanNet, ShitiNet) to enrich diversity and ensure coverage of underrepresented scenarios, such as primary and career education. Furthermore, guided by the Catalogue of Undergraduate Programs in Regular Higher Education Institutions [1] issued by China's Ministry of Education, we curated a large body of review materials and exam questions across 13 academic disciplines, including philosophy, education, law, literature, history, science, engineering, agriculture, medicine, military science, management, and the arts. This effort significantly improves distributional balance and provides a more faithful reflection of real applications.

*Manual collection of private data.* Data contamination remains one of the most critical challenges in constructing evaluation datasets for LLMs. To mitigate this risk, we manually collected additional data from private resources, such as internal school exam papers. Unlike widely circulated national exams, these materials have never appeared on the public Internet or been included in large-scale web crawls, effectively reducing the risk of leakage. Incorporating such private data enhances the reliability and fairness of the benchmark, while providing a more rigorous assessment of models.

*LLM-generated data.* Given the difficulty of directly obtaining data in the cultivation dimension, we leveraged LLMs to generate a substantial number of scenario-based question–answer pairs, aiming to supplement gaps in existing resources. To ensure the quality of the synthetic data, we invited five

---

[1] http://www.moe.gov.cn/srcsite/A08/moe_1034/s4930/202403/
W020240319305498791768.pdf

Table 1: Statistics of OmniEduBench and more detailed per-subject information are shown in the Appendix. Bilingual names and abbreviations of six knowledge and six cultivation dimensions.

| *Knowledge dimension* | | | *Cultivation dimension* | | |
|---|---|---|---|---|---|
| English name | Abbreviation | Chinese name | English name | Abbreviation | Chinese name |
| Law & Politics | LP | 法律与政治 | Character & Values | CV | 品格与价值观 |
| Foundational Disciplines | FD | 基础学科 | Personalized Development | PD | 个性化发展 |
| Humanities & History | HH | 人文与历史 | Social & Interpersonal Skills | SIS | 社会与人际交往 |
| Medicine & Health | MH | 医学与健康 | Thinking & Cognitive Skills | TCS | 思维与认知能力 |
| Interdisciplinary & Integrated Subjects | IIS | 综合与交叉学科 | Teaching Feedback & Support | TFS | 教学反馈与支持) |
| Social Sciences & Economics Management | SSEM | 社会科学与经济管理 | Emotional & Mental Health | EMH | 情感与心理健康 |

| Category | Subjects | Questions | Category | Subjects | Questions | Category | Subjects | Questions |
|---|---|---|---|---|---|---|---|---|
| *In terms of dimension* | | | *In terms of Knowledge* | | | *In terms of Cultivation* | | |
| Knowledge | 41 | 18,121 | LP | 4 | 1,455 | CV | 2 | 694 |
| Cultivation | 20 | 6,481 | FD | 11 | 7,918 | PD | 3 | 1,031 |
| *In terms of different level* | | | HH | 10 | 5,331 | SIS | 3 | 736 |
| K-12 Schools | 10 | 4,384 | MH | 3 | 918 | TCS | 6 | 1,900 |
| High school | 11 | 6,735 | IIS | 4 | 914 | TFS | 1 | 193 |
| College | 30 | 6,364 | SSEM | 9 | 1,643 | EMH | 5 | 1,833 |
| Total | 61 | 24,602 | Total | 41 | 18,179 | Total | 20 | 6,387 |

Table 2: Comparison of OmniEduBench with existing benchmarks. Here, Mixed (X) indicates X types of mixed question types, and General (K) signifies that the dataset covers all subjects and educational stages with K categories. ZH and EN represent Chinese and English, respectively.

| Benchmark | Dimension | Size | Type | Data source | Domain | Subject | Language |
|---|---|---|---|---|---|---|---|
| C-Eval (Huang et al., 2023) | Knowledge | 13.9K | Multiple answer | Publicly available | General (4) | 52 | ZH |
| EduBench (Xu et al., 2025) | Knowledge | 18.8K | Multiple choice | LLM-generated | General (9) | 9 | ZH & EN |
| CMATH (Wei et al., 2023) | Knowledge | 1.7K | Multiple choice | Publicly available | Math | 1 | ZH |
| E-EVAL Hou et al. (2024) | Knowledge | 4.35K | Multiple answer | Publicly available | K-12 | 23 | ZH |
| GAOKAO (Zhang et al., 2023) | Knowledge | 2.8K | Mixed (7) | Publicly available | High school | 9 | ZH |
| GSM8K (Cobbe et al., 2021) | Knowledge | 8.5K | Multiple choice | Publicly available | Math | 1 | EN |
| MATH (Hendrycks et al., 2021b) | Knowledge | 12.5K | Multiple choice | Publicly available | Math | 7 | EN |
| OmniEduBench (ours) | Knowledge & Cultivation | 24.6K | Mixed (11) | Publicly & Private & LLM | General (12) | 61 | ZH |

education experts to conduct discussions on 20 cultivation subjects and consulted relevant books, papers, and other materials. The collected content was organized into a database, which was then provided to the LLM to enhance the fidelity and accuracy of the generated data. For generated questions, to increase their challenge, we designed highly confounding distractors via prompts and conducted sampling checks and revisions with expert verification (please see more details in expert verification). Finally, we collected a total of 927K question–answer (Q&A) pairs, including 21K from publicly available data, 106K from private data, and 800K generated by LLMs.

**Dataset cleaning.** The entire data cleaning process consists of multiple steps. (1) We used MinerU (Wang et al., 2024) to convert the collected 927K Q&A pairs into Markdown (md) format, enabling structured management and efficient information extraction. (2) Detailed metadata were extracted for each question, including subject, grade level, question type, and knowledge tags, to construct comprehensive question profiles that facilitate data management and subsequent analysis. (3) Standard data cleaning procedures were applied, including deduplication, removal of questions with missing key content, filtering of sensitive or inappropriate content, and exclusion of questions that rely on external information. After the cleaning process, we obtained a total of 657K Q&A pairs.

**Dual-machine filtering.** To ensure OmniEduBench is a high-quality and challenging benchmark, we implemented a dual-model filtering mechanism on an initial set of 657K Q&A pairs. Specifically, we first evaluated all questions using QWQ32B (Team, 2025c), retaining only those that the model answered incorrectly. This initial filtering resulted in a subset of 430K Q&A pairs. These questions then underwent a second filtering stage with the same strategy, this time using Qwen3-235B (Yang et al., 2025), ultimately yielding the final set of 50K high-quality and challenging data.

**Expert verification.** We recruited 50 master's students to perform an initial quality check on the dataset based on five predefined dimensions (in Table 3), removing any data that did not meet the criteria, which resulted in a final set of 24.602K Q&A pairs for OmniEduBench. Among these 24.602K Q&A pairs, 26.1% were sourced from publicly available data, 28.1% from private data, and the remaining 25.8% from LLM-generated data. Subsequently, we invited 5 senior verification experts to conduct a rigorous quality review on a 15% random sample drawn separately from publicly available, private, and LLM-generated data of OmniEduBench. The review results in Table 3, indicate that the dataset maintains high overall quality, demonstrating both reliability and applicability.

Table 3: Expert validation results for the OmniEduBench dataset.

| Metric English name | Metric Chinese name | Average | Standard deviation | Inter-rater agreement |
|---|---|---|---|---|
| Overall quality | 整体质量 | 4.8 | 0.1 | 0.90 |
| Clarity | 问题清晰度 | 4.5 | 0.2 | 0.85 |
| Option perplexity | 选项困惑度 | 4.8 | 0.3 | 0.83 |
| Accuracy | 答案准确性 | 4.8 | 0.1 | 0.90 |
| Cultivation value | 育人价值 | 4.6 | 0.2 | 0.88 |

下列说法错误的是：
The following statements are incorrect:

A. 甲苯与甘油燃烧时产生的水量在等物质的量条件下保持恒定
A. The amount of water produced by the combustion of toluene and glycerol remains constant under the condition of equal amounts

B. 溴乙烷发生消去反应后，可用酸性高锰酸钾溶液直接检验生成的乙烯
B. After the elimination reaction of ethyl bromide, the ethylene produced can be directly tested with acidic potassium permanganate solution

C. 醛类物质在碱性条件下与银氨溶液共热可发生银镜反应
C. Aldehydes can undergo silver mirror reaction when co-heated with silver ammonia solution under alkaline conditions

D. 碳酸钠溶液可溶解苯酚生成苯酚钠和碳酸氢钠
D. Sodium carbonate solution dissolves phenol to produce sodium phenol and sodium bicarbonate

答案：B
Answer: B

(a) Single-choice question type within the knowledge dimension

有学生在参观烈士陵园时嬉笑打闹，我很生气，该怎么处理？
Some students laughed and played while visiting the martyrs' cemetery, I feel very angry, how should I deal with it?

A. 当场严厉批评，让他道歉并写检讨。
A. Criticize harshly on the spot, ask him to apologize and write a review.

B. 暂停他的课外活动资格一周作为惩罚。
B. Suspension of his extracurricular activities for one week as punishment.

C. 课后找他谈话：你知道我们为什么要来这里吗？你觉得这样的行为对得起先烈吗？你希望别人怎么纪念你？
C. Talk to him after class: Do you know why we are here? Do you think such behavior is worthy of the martyrs? How do you want others to remember you?

D. 让他下次参观时担任讲解员，负责介绍一位烈士的事迹。
D. Let him serve as a docent on his next visit, responsible for introducing the deeds of a martyr.

答案：D
Answer: D

(b) Single-choice question type within the cultivation dimension

Figure 3: Example of (a) a single-choice question in the knowledge from a college chemist. (b) A single-choice question in the cultivation. English translations are shown for better readability.

## 3.3 EVALUATION CRITERIA

Based on the characteristics of different question types, we adopt two evaluation metrics: (1) **Choice**. For questions with a standard answer, we directly evaluate the provided answer. This simplifies the scoring process, as the model only needs to select the most appropriate option, thereby reducing ambiguity in assessment. (2) **LLM-assisted scoring**. For short-answer questions that may have multiple valid forms but are semantically equivalent, we employ an LLM-assisted scoring method. This approach provides greater flexibility, avoids imposing unnecessary constraints on the model, and allows for a more accurate evaluation of the model's semantic understanding and expression.

## 3.4 STATISTICS

Through rigorous data filtering and expert validation, we collected 18.121K high-quality question–answer pairs for the knowledge and 6.481K for the cultivation. As illustrated in Figure 1 and summarized in Table 1, with more detailed per-subject statistics provided in the Appendix, the dataset spans 12 major categories, as shown in Table 1, including K-12, higher school, university-level courses, and cultivation aspects such as emotion and reasoning, covering a total of 61 specific scenarios. Figures 3 and 4 present some representative examples in different dimensions and question types. The questions exhibit wide variability in type and difficulty and are sourced from diverse origins, primarily newly collected from public or private resources or manually constructed.

## 4 EXPERIMENTS

In this section, we evaluate the performance of state-of-the-art (SOTA) methods in both English and Chinese. The experimental results indicate that OmniEduBench remains a competitive benchmark.

## 4.1 EXPERIMENTAL SETUP

**Baselines.** We evaluate 11 mainstream large language models (LLMs) in total, including 3 cutting-edge closed-source models and 8 open-source models, one of which is a newly released education-oriented model. The closed-source models are GPT-4o (Hurst et al., 2024), Gemini-2.5 Pro (Comanici et al., 2025), and Claude-4 Sonnet (Team, 2025a). For the open-source models, we consider two main factors. First, they are grouped by parameter size into small (8B), medium (14B/32B/36B),

BAT（棕色脂肪组织）细胞膜上具有下列哪些受体？（多选）
Which of the following receptors are present on the BAT (brown adipose tissue) cell membrane? (Multiple Choice)

A. 葡萄糖受体
A. Glucose receptors

B. 神经递质受体
B. Neurotransmitter receptors

C. 促胰液素受体
C. Secretin-stimulating receptors

D. 甲状腺激素受体
D. Thyroid hormone receptors

答案：B；C
Answer: B; C

(a) Multiple-choice question within the knowledge dimension

已知抛物线 Γ：y² = 4x的焦点为F，过F作斜率为k的直线交 Γ 于A、B两点。（1）若|AB|=10，求线段AB中点的横坐标；（2）证明：在抛物线 Γ 上所有的点中，顶点到焦点F的距离最小；
It is known that the focus of the parabolic Γ: y²=4x is F, and the straight line with a slope of k crosses F Γ at points A and B. (1) If | AB |=10, find the abscissa of the midpoint of the line segment AB. (2) Prove: Among all the points on the parabolic Γ, the distance from the vertex to the focal point F is the smallest.

答案：（1）AB中点横坐标为4；（2）顶点到焦点距离最小；
Answer: (1) The abscissa of the midpoint of AB is 4. (2) The distance from the vertex to the focal point is the smallest.

(b) Short-answer question within the knowledge dimension

Figure 4: Example of (a) a multiple-choice question in the knowledge from Biology. (b) A short-answer question in the knowledge from Math. English translations are shown for better readability.

Table 4: Zero-shot average accuracy (%) across six categories in the **knowledge**. The highest accuracy is **bold**, and the second highest is underlined. More results are provided in the Appendix.

| Model | Parameters | Access | Creator | FD | HH | SSEM | LP | MH | IIS | Average |
|---|---|---|---|---|---|---|---|---|---|---|
| Qwen3 | 8B | Weights | Alibaba | 53.02 | 38.53 | 36.58 | 30.17 | 36.71 | 37.75 | 43.86 |
| Qwen3 | 14B | Weights | Alibaba | 36.32 | 36.78 | 35.12 | 27.29 | 36.82 | 35.67 | 35.62 |
| MuduoLLM | 14B | Weights | BNU & TAL | 28.20 | 40.82 | 32.99 | 36.15 | 39.11 | 31.40 | 33.68 |
| QwQ | 32B | Weights | Alibaba | 61.25 | 48.51 | 42.24 | 49.90 | 55.01 | 47.26 | 53.87 |
| Seed-OSS | 36B | Weights | ByteDance | 48.81 | 50.14 | 45.34 | 48.66 | 61.00 | 49.56 | 49.53 |
| Qwen2.5 | 72B | Weights | Alibaba | 19.53 | 30.95 | 20.57 | 13.26 | 23.86 | 20.90 | 22.76 |
| Qwen3 | 235B (22B active) | Weights | Alibaba | 34.24 | 47.01 | 36.21 | 44.26 | 58.71 | 46.61 | 40.82 |
| DeepSeek-V3.1 | 671B (37B active) | Weights | DeepSeek | 31.65 | 40.65 | 35.00 | 29.42 | 50.54 | 45.19 | 36.05 |
| HuatuoGPT-o1_ans (Medicine) | 7B | Weights | CUHK-Shenzhen | 23.67 | 24.23 | 23.38 | 18.79 | 29.74 | 20.94 | 23.46 |
| Qwen2.5-Math-instruct_ans (Math) | 7B | Weights | Alibaba | 33.07 | 8.43 | 11.51 | 8.39 | 10.67 | 13.93 | 14.33 |
| GPT-4o | Undisclosed | API | OpenAI | 21.15 | 26.94 | 23.92 | 22.13 | 34.75 | 27.13 | 24.17 |
| Claude-4 Sonnet | Undisclosed | API | Anthropic | 41.49 | 44.29 | 35.36 | 27.56 | 34.86 | 42.34 | 40.35 |
| ERNIE-4.5-Turbo_ans | Undisclosed | API | BaiDu | 39.89 | 51.59 | 45.61 | 52.07 | 59.15 | 47.43 | 49.29 |
| Spark_Pro_ans | Undisclosed | API | iFlytek | 22.96 | 37.03 | 41.58 | 41.03 | 36.82 | 37.90 | 36.22 |
| Gemini-2.5 Pro | Undisclosed | API | Google | 73.83 | 55.13 | 46.68 | 55.40 | 60.68 | 54.16 | 62.76 |

and large (72B/235B/671B) scales. Second, they are categorized by functionality into: (a) general instruction-following models (Qwen2.5 (Yang et al., 2024), Qwen3 (Yang et al., 2025)); (b) general reasoning models (QwQ (Team, 2025c), Seed-OSS (Team, 2025b), DeepSeek-V3.1 (Liu et al., 2024b)); and (c) education-specific models (MuduoLLM (from BNU & TAL, 2025)).

**Implementation details.** In our experimental setup, we evaluate all large language models under both zero-shot and few-shot settings, with few-shot examples (0-, 1-, 3-, and 5-shot) drawn from a separately partitioned development set, distinct from the evaluation set. All open-source models are run using their official code, while closed-source models are accessed via official APIs. We consistently use Gemini-2.5 Pro as the LLM-assisted scoring model, unless otherwise specified. Following C-Eval (Huang et al., 2023) and GSM8K (Cobbe et al., 2021), we set temperature (T) to 0 during inference, which corresponds to using greedy decoding. At each generation step, the model selects the token with the highest probability, ensuring deterministic outputs that minimize randomness and produce the most common and stable answers. We conducted three independent runs and averaged the results. Due to T=0, results across three runs showed negligible differences.

Table 5: Zero-shot average accuracy (%) across six categories in the **cultivation**. The highest accuracy is **bold**, and the second highest is underlined. More results are provided in the Appendix.

| Model | Parameters | Access | Creator | TCS | EMH | SIS | CV | PD | TFS | Average |
|---|---|---|---|---|---|---|---|---|---|---|
| Qwen3 | 8B | Weights | Alibaba | 70.95 | 66.67 | 69.16 | 62.25 | 70.13 | 77.20 | 68.62 |
| Qwen3 | 14B | Weights | Alibaba | 67.79 | 60.77 | 63.72 | 56.20 | 64.31 | 71.50 | 63.60 |
| MuduoLLM | 14B | Weights | BNU & TAL | 64.42 | 60.77 | 63.45 | 67.51 | 64.77 | 63.96 |
| QwQ | 32B | Weights | Alibaba | 73.16 | 68.36 | 69.84 | 65.13 | 71.77 | 72.02 | 70.27 |
| Seed-OSS | 36B | Weights | ByteDance | 70.74 | 65.30 | 66.03 | 62.82 | 67.12 | 70.47 | 67.18 |
| Qwen2.5 | 72B | Weights | Alibaba | 67.89 | 64.38 | 65.62 | 59.51 | 65.57 | 67.88 | 65.34 |
| Qwen3 | 235B (22B active) | Weights | Alibaba | 67.84 | 61.10 | 64.54 | 55.76 | 64.40 | 70.47 | 63.74 |
| DeepSeek-V3.1 | 671B (37B active) | Weights | DeepSeek | 71.58 | 65.41 | 69.02 | 61.96 | 71.00 | 77.20 | 68.55 |
| HuatuoGPT-o1_ans (Medicine) | 7B | Weights | CUHK-Shenzhen | 42.58 | 45.57 | 48.10 | 41.79 | 43.65 | 52.85 | 44.44 |
| Qwen2.5-Math-instruct_ans (Math) | 7B | Weights | Alibaba | 28.74 | 31.44 | 20.52 | 27.52 | 16.97 | 22.28 | 26.34 |
| GPT-4o | Undisclosed | API | OpenAI | 61.63 | 59.57 | 59.24 | 55.33 | 57.71 | 65.80 | 59.57 |
| ERNIE-4.5-Turbo_ans | Undisclosed | API | BaiDu | 70.11 | 62.85 | 66.30 | 56.77 | 67.02 | 70.47 | 65.65 |
| Spark_Pro_ans | Undisclosed | API | iFlytek | 68.05 | 61.87 | 64.95 | 61.10 | 67.80 | 72.02 | 65.24 |
| Claude-4-sonnet | Undisclosed | API | Anthropic | 71.95 | 70.05 | 70.92 | 64.55 | 69.25 | 71.50 | 70.03 |
| Gemini-2.5-pro | Undisclosed | API | Google | 72.26 | 66.07 | 70.79 | 65.71 | 70.32 | 67.36 | 69.14 |

Table 6: Average accuracy (%) across six categories in one-shot, three-shot, and five-shot settings for the knowledge dimension. The highest accuracy is **bold**, and the second highest is underlined.

| Model | Parameters | Access | Creator | FD | HH | SSEM | LP | MH | IIS | Average |
|-------|-----------|--------|---------|-----|-----|------|-----|-----|------|---------|
| *One-shot setting* | | | | | | | | | | |
| Qwen3 | 8B | Weights | Alibaba | **52.80** | 46.45 | **41.90** | 29.76 | 40.20 | 40.59 | **41.95** |
| MuduoLLM | 14B | Weights | BNU & TAL | 27.36 | 47.79 | 36.72 | 34.98 | 40.74 | 34.35 | 36.99 |
| Qwen2.5 | 72B | Weights | Alibaba | 21.42 | 40.43 | 27.04 | 20.96 | 28.10 | 22.65 | 26.77 |
| Qwen3 | 235B (22B activate) | Weights | Alibaba | 37.72 | **60.79** | 44.03 | **45.77** | **59.59** | **54.05** | **50.12** |
| DeepSeek-V3.1 | 671B (37B activate) | Weights | DeepSeek | 30.00 | 41.73 | 34.65 | 30.72 | 49.67 | 42.12 | 38.15 |
| *Three-shot setting* | | | | | | | | | | |
| Qwen3 | 8B | Weights | Alibaba | **52.98** | 46.00 | 39.74 | 30.65 | 39.32 | 40.85 | 41.59 |
| MuduoLLM | 14B | Weights | BNU & TAL | 27.32 | 46.86 | 35.79 | 33.81 | 39.54 | 32.42 | 35.96 |
| Qwen2.5 | 72B | Weights | Alibaba | 21.43 | 40.86 | 27.27 | 20.41 | 27.12 | 23.88 | 26.83 |
| Qwen3 | 235B (22B activate) | Weights | Alibaba | 37.52 | **60.70** | 43.40 | **45.77** | **59.48** | **52.57** | **49.54** |
| DeepSeek-V3.1 | 671B (37B activate) | Weights | DeepSeek | 29.09 | 41.42 | 34.02 | 28.59 | 48.80 | 42.06 | 37.33 |
| *Five-shot setting* | | | | | | | | | | |
| Qwen3 | 8B | Weights | Alibaba | **56.86** | 46.70 | 39.44 | 30.65 | 38.24 | 42.23 | 42.35 |
| MuduoLLM | 14B | Weights | BNU & TAL | 26.93 | 46.57 | 36.28 | 35.74 | 39.11 | 34.57 | 36.53 |
| Qwen2.5 | 72B | Weights | Alibaba | 21.46 | 41.19 | 26.96 | 20.82 | 26.03 | 28.56 | 27.50 |
| Qwen3 | 235B (22B activate) | Weights | Alibaba | 37.40 | **60.41** | **44.13** | **45.77** | **58.61** | **55.58** | **50.32** |
| DeepSeek-V3.1 | 671B (37B activate) | Weights | DeepSeek | 29.39 | 41.17 | 32.87 | 28.45 | 47.49 | 38.95 | 36.39 |

## 4.2 MAIN RESULTS

We evaluated all baseline models on OmniEduBench, reporting both per-task category and overall accuracy, as shown in Tables 4 and 5). Results show that in the knowledge dimension, Gemini-2.5 Pro achieves the highest accuracy at 62.78%, while in the cultivation dimension, the reasoning-enhanced version of QWQ performs best with an accuracy of 70.27%. This performance highlights the challenging nature and strong discriminative power of the constructed OmniEduBench.

In the knowledge dimension, it is evident that, except for Gemini-2.5 Pro, closed-source models generally perform worse than open-source models on our OmniEduBench. For example, GPT-4o achieves an accuracy of 24.17%, far below that of Qwen3-8B. This may indicate that the GPT series has relatively weak robustness when handling Chinese education exam-style questions. Meanwhile, model architecture has a significant impact on performance, such as Seed-OOS outperforms the Qwen family by more than 10%. In the cultivation dimension, models generally perform better than in the knowledge dimension, which may be due to the fact that the cultivation tasks mainly consist of multiple-choice questions, making them simpler compared to knowledge tasks with 11 common exam question types. However, differences in performance between different model architectures still exist. Overall, GPT-4o performs the worst in both dimensions, with accuracy largely concentrated around 59.57%, possibly because it has not been specifically optimized for this dimension.

## 4.3 ANALYSIS AND FINDINGS

In this section, we further conduct extensive experiments at multiple levels, including few-shot examples, OmniEduBench HARD, and various LLM-assisted scoring methods.

**Results in few-shot examples.** In Table 6, we present in-context experimental results using different numbers of shots. As the number of shots increases, model performance generally improves; however, the overall gain is limited when considering the average results. We speculate that the drop in accuracy for some models is due to the fact that they have not (or not appropriately) incorporated few-shot examples during the instruction tuning stage. These findings suggest that while few-shot prompting can be beneficial for certain models, its effectiveness strongly depends on the model's pretraining and instruction tuning strategies. Moreover, the limited average improvement indicates that simply increasing the number of shots may not always lead to substantial gains, highlighting the need for more sophisticated methods to integrate few-shot examples effectively.

**Results on OmniEduBench HARD.** In Figures 5 and 6, we present the average accuracy of each model on OmniEduBench HARD. The OmniEduBench HARD subset consists of the bottom 26% of samples based on the performance of 11 evaluated models, totaling approximately 1.552K cultivation samples and 7.620K knowledge samples (9.172K examples in total). To ensure the subset reflects true difficulty rather than model-specific weaknesses, we invited 10 master's students to

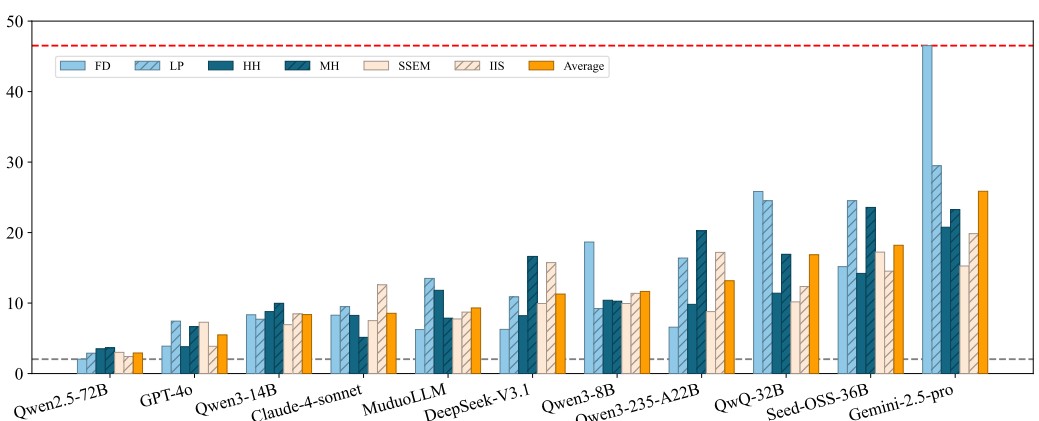

Figure 5: Zero-shot average accuracy (%) on the knowledge dimension of OmniEduBench HARD.

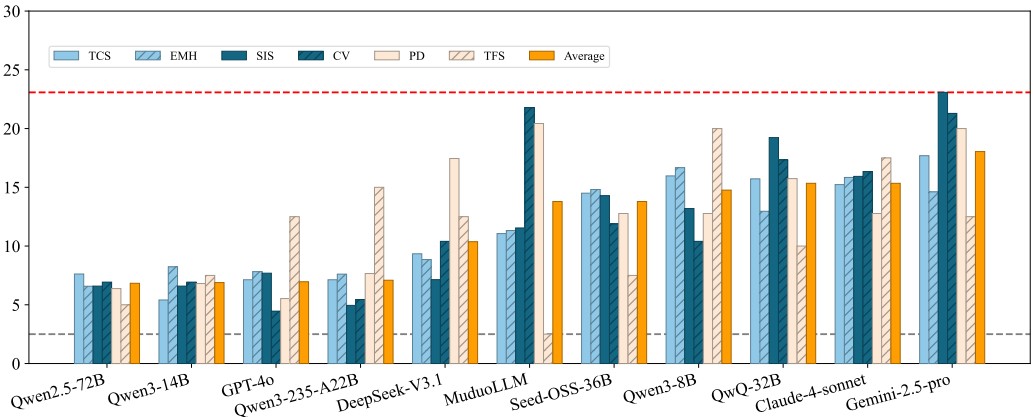

Figure 6: Zero-shot average accuracy (%) on the cultivation dimension of OmniEduBench HARD.

manually review these samples, confirming that the problems in the HARD subset generally align with higher human-perceived difficulty. The experimental results show that: (1) all 11 LLMs exhibit a significant performance drop on OmniEduBench HARD, with even the best-performing model, Gemini, achieving less than 50% accuracy; (2) Qwen2.5-72 performs the worst, significantly lower than the other models, indicating limited capability in handling difficult samples. These findings indicate that further research is needed to enhance LLMs' ability to generalize and maintain high performance on hard subsets of educational benchmarks.

**Results using different LLM-assisted scoring methods.** In Table 7, we present the experimental results using different LLM-assisted scoring methods. The performance of the scoring model directly affects the evaluation outcomes: higher-quality scoring models provide more accurate assessments, leading to more precise measurements of the evaluated models' capabilities. In this study, we employed three scoring models of varying quality. Overall, GPT-4o performed relatively poorly as a scoring model, failing to accurately evaluate the responses of LLMs. Consequently, the overall effectiveness of LLM-assisted evaluation is reduced when GPT-4o is used, highlighting the critical importance of selecting high-quality scoring models to ensure accurate and meaningful assessments. These findings suggest that the choice of scoring model can substantially influence the perceived performance of evaluated LLMs, and careful selection of scoring models is necessary.

## 5 CONCLUSIONS, DISCUSSIONS AND LIMITATIONS

In this paper, we present OmniEduBench, a comprehensive Chinese educational benchmark designed to address the limitations of existing Chinese educational evaluation benchmarks. By moving beyond simple knowledge retrieval, the benchmark provides a holistic assessment of LLMs' capabilities across two core dimensions: the knowledge and cultivation dimensions. We conducted extensive experiments on 11 mainstream LLMs, revealing significant performance gaps. While

Table 7: Zero-shot average accuracy (%) across six categories in the knowledge using different LLM-assisted scoring methods. The highest accuracy is **bold**, and the second highest is underlined.

| Model | Parameters | Access | Creator | FD | HH | SSEM | LP | MH | IIS | Average |
|---|---|---|---|---|---|---|---|---|---|---|
| *Qwen3-A235B-assisted scoring method (Yang et al., 2025)* | | | | | | | | | | |
| QwQ | 32B | Weights | Alibaba | 61.26 | 55.82 | 45.22 | 50.10 | 55.66 | 48.36 | 56.39 |
| Seed-OSS | 36B | Weights | ByteDance | 51.16 | 63.04 | 51.55 | 50.38 | **62.31** | 57.33 | 55.49 |
| GPT-4o | Undisclosed | API | OpenAI | 23.59 | 36.77 | 28.79 | 23.92 | 35.73 | 31.40 | 28.96 |
| Claude-4 Sonnet | Undisclosed | API | Anthropic | 43.54 | 55.52 | 40.47 | 27.77 | 36.17 | 47.48 | 45.34 |
| Gemini-2.5 Pro | Undisclosed | API | Google | **75.01** | **65.67** | **52.95** | **56.29** | 61.44 | **61.49** | **67.41** |
| *Gemini-2.5 Pro-assisted scoring method (Comanici et al., 2025)* | | | | | | | | | | |
| QwQ | 32B | Weights | Alibaba | 61.25 | 48.51 | 42.24 | 49.90 | 55.01 | 47.26 | 53.87 |
| Seed-OSS | 36B | Weights | ByteDance | 48.81 | 50.14 | 45.34 | 48.66 | **61.00** | 49.56 | 49.53 |
| GPT-4o | Undisclosed | API | OpenAI | 21.15 | 26.94 | 23.92 | 22.13 | 34.75 | 27.13 | 24.17 |
| Claude-4 Sonnet | Undisclosed | API | Anthropic | 41.49 | 44.29 | 35.36 | 27.56 | 34.86 | 42.34 | 40.35 |
| Gemini-2.5 Pro | Undisclosed | API | Google | **73.83** | **55.13** | **46.68** | **55.40** | 60.68 | **54.16** | **62.76** |
| *GPT-4o-assisted scoring method (Hurst et al., 2024)* | | | | | | | | | | |
| QwQ | 32B | Weights | Alibaba | 56.61 | 42.87 | 37.86 | 49.28 | 54.58 | 37.97 | 49.26 |
| Seed-OSS | 36B | Weights | ByteDance | 45.07 | 47.89 | 41.94 | 48.52 | **60.78** | 43.54 | 46.61 |
| GPT-4o | Undisclosed | API | OpenAI | 20.38 | 23.78 | 22.03 | 22.06 | 34.31 | 23.30 | 22.51 |
| Claude-4 Sonnet | Undisclosed | API | Anthropic | 40.43 | 41.70 | 31.89 | 27.90 | 35.08 | 34.79 | 38.48 |
| Gemini-2.5 Pro | Undisclosed | API | Google | **70.15** | **51.38** | **44.13** | **55.88** | 60.57 | **46.83** | **59.49** |

some models performed well on the knowledge dimension, their performance on cultivation tasks dropped substantially, with even the best-performing models trailing human-level performance by nearly 30%. These findings indicate that despite recent advancements in LLM technology, current models still lack the deep reasoning and pedagogical skills necessary to function effectively as educational assistants. We believe OmniEduBench will serve as an important tool for guiding future research. Looking ahead, OmniEduBench plans to explore more complex question types in the cultivation dimension and introduce multimodal educational scenarios, further enhancing the benchmark's role in evaluating and guiding the comprehensive capabilities of LLMs and MLLMs.

**Ethics statement.** Our constructed OmniEduBench educational benchmark is built from publicly available educational resources as well as authorized private resources permitting open-source use, strictly adhering to copyright and licensing requirements. All data have been systematically processed to remove personally identifiable information (PII) and sensitive content, ensuring privacy and security. The dataset is intended solely for research purposes, aiming to advance the development and evaluation of large language models (LLMs) in educational scenarios.

**Reprodicibility statement.** To ensure reproducibility, we provide detailed descriptions of the dataset construction process, annotation criteria, and experimental settings in both the main paper and the Appendix. The proposed OmniEduBench education dataset, together with preprocessing scripts, evaluation metrics, and model prompts, will be publicly released upon acceptance. All experiments were conducted using standard LLM APIs or open-source checkpoints, with model versions, hyperparameters, and evaluation protocols explicitly documented. This ensures that other researchers can faithfully replicate our results and readily extend the benchmark in future studies.

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

## A SUPPLEMENTARY MATERIAL

**Use of LLMs.** In this paper, LLMs were utilized in two primary ways: (1) as auxiliary tools for data cleaning and preliminary quality checks under human supervision. (2) As evaluation targets in benchmark experiments. To ensure data quality, no content directly generated by LLMs was included in the released dataset. During manuscript preparation, LLMs were employed for minor language polishing. All ideas, methodologies, and conclusions are original contributions of authors.

## A.1 SUPPLEMENTARY STATISTICS OF OMNIEDUBENCH

In Table 8, we present the bilingual names and abbreviations of all subjects in the knowledge dimension. In Table 9, we present the bilingual names and abbreviations of all subjects in the cultivation dimension. In Tables 10, 11, and 12, we present the detailed data distribution for all 61 subjects.

In Figures 7 and 8, more examples of various questions in the knowledge and cultivation dimensions.

名词解释：大晟词派
Definition:Dasheng School of Ci Poetry:

答案：词是音乐文学，词风常随乐曲的推移而产生变化，北宋后期音乐机构大晟词府的建立，促进了当时词风的变化，有以周邦彦为代表的大晟词派的产生。周邦彦作词以协律为主，长调尤尚铺叙，于音律和谐中求词句的浑雅，极沉郁顿挫之致，北宋的慢词发展到周邦彦的清真词，以知音律、备法度和风格醇雅著称，既无柳永的'词语尘下'之病，又无苏轼的'多不协律'之讥，成为后世词人取法的榜样，影响极为深远。

Answer: The establishment of the Dasheng Mansion, a music institution in the late Northern Song Dynasty, promoted the change of the style of words at that time, and the emergence of the Dasheng Poetry School represented by Zhou Bangyan. Zhou Bangyan's lyrics are mainly based on harmony, and the long tone is especially laid out, seeking the elegance of the words and sentences in the harmony of the rhythm, extremely melancholic and frustrated, the slow words of the Northern Song Dynasty developed to Zhou Bangyan's halal words, known for knowing the rhythm, preparing the law and the style of mellow elegance, neither Liu Yong's 'words under the dust', nor Su Shi's ridicule of 'more incongruity', which has become a model for later generations of poets to learn from, and the influence is extremely far-reaching.

Noun explanation question within the knowledge dimension

《西厢记》的戏剧冲突线索和情节发展
The dramatic conflict clues and plot development of "The West Chamber"

答案：西厢记》的戏剧冲突有两条线索一是以老夫人及郑恒为一方，以崔莺莺、张生、红娘为另一方的冲突二是崔莺莺、张生、红娘之间的矛盾冲突。前者是主线，后者是辅线。两条线索互相制约，交错展开。第二本中孙飞虎兵围普救寺是一个大关目。在此之前，莺莺、张生与老夫人的矛盾冲突是潜在的·表现为崔、张一见钟情之后，两人相互爱慕、希望接近，与老夫人治家谨严、冰霜之操之间的矛盾，也就是莺莺对爱情的渴望与老夫人维护礼教、维护门阀体面之间的无形冲突。孙飞虎的出现，使剧情的发展顿时激化。老夫人的许愿，张生的退敌，崔、张二人以为天遂人愿，不料老夫人变卦，使他们的理想领成泡影。此后，戏剧冲突发生了一系列变化崔、张与老夫人的冲突表面化，并且，由于对老夫人的极端不满，他们私下以'非法'的形式结合了。'拷红'是戏剧冲突的又一大转折。崔、张私下结合被老夫人发觉，于是拷问红娘，老夫人与红娘之间发生直接冲突，红娘抓住老夫人的弱点，勇敢斗争，迫使老夫人不得不承认崔、张的关系，但矛盾并没有就此解决。老夫人在承认了崔、张的婚事后，立即提出附加条件张生必须应考得官，才能成就婚事，这使得戏剧冲突再起波澜。张生答应老夫人的条件，崔、张被迫分离，但崔莺莺却明确提出此一去得官不得官，疾便回来'。老夫人坚持相府不招白衣女婿，莺莺则认为，'但得一个并头莲，强似状元及第二人物之间的冲突，已经明显表现为两种婚姻观念的冲突。第五本是戏剧冲突的最后解决。尽管郑恒竭力破坏，张生仍然在得官归来后与莺莺成婚，使故事得以大团圆结局。

Answer: There are two clues to the dramatic conflict in "The West Chamber": one is the conflict between the old lady and Zheng Heng on one side and Cui Yingying, Zhang Sheng, and the matchmaker on the other, and the other is the conflict between Cui Yingying, Zhang Sheng, and the matchmaker. The former is the main line, and the latter is the auxiliary line. The two clues restrict each other and unfold in a staggered manner. In the second book, Sun Feihu's siege of the Pujiu Temple is a big pass. Prior to this, the contradictions and conflicts between Yingying, Zhang Sheng and the old lady were latent After falling in love with Cui and Zhang at first sight, the two fell in love with each other and hoped to get closer, and the contradiction between the old lady's rigorous management of the family and the frosty conduct, that is, the invisible conflict between Yingying's desire for love and the old lady's maintenance of etiquette and the decency of the gatekeeper. The appearance of Sun Feihu suddenly intensified the development of the plot. The old lady's wish, Zhang Sheng's retreat from the enemy, Cui and Zhang thought that heaven had fulfilled their wishes, but the old lady changed her mind and their ideals came to naught. Since then, a series of theatrical conflicts have taken place, and the conflict between Cui, Zhang and the old lady has surfaced, and, due to extreme dissatisfaction with the old lady, they have been united in private in an 'illegal' form. 'Torture Red' is another major turning point in the dramatic conflict. Cui and Zhang's private union was discovered by the old lady, so they tortured the matchmaker, and there was a direct conflict between the old lady and the matchmaker. The matchmaker seized the old lady's weakness and fought bravely, forcing the old lady to admit the relationship between Cui and Zhang. But the contradiction was not resolved. After the old lady admitted the marriage of Cui and Zhang, she immediately put forward additional conditions: Zhang Sheng must be an official in order to achieve the marriage, which made the dramatic conflict make waves again. Zhang Sheng agreed to the old lady's conditions, and Cui and Zhang were forced to separate. But Cui Yingying clearly proposed that 'if you go to an official position this time, you will come back when you are sick'. The old lady insisted that the prime minister's mansion did not recruit a son-in-law in white, and Yingying believed: 'But there is a parallel lotus, which is stronger than the conflict between the champion and the second person, which has obviously manifested itself as a conflict between the two marriage concepts. The fifth book is the final resolution of the dramatic conflict. Despite Zheng Heng's efforts to destroy it, Zhang Sheng still married Yingying after returning from the official position, so that the story could end happily.

Essay Question within the knowledge dimension

Figure 7: Examples of different questions in the knowledge dimension and cultivation dimensions.

Table 8: Bilingual names and abbreviations of all subject in the knowledge dimension.

| Abbreviation | English Name | Chinese Name |
| --- | --- | --- |
| MATH | Mathematics | 数学 |
| CHEM | Chemistry | 化学 |
| BIO | Biology | 生物 |
| PHY | Physics | 物理 |
| NSCI | Nature & Science | 自然与科学 |
| PSTAT | Probability & Statistics | 概率论与数理统计 |
| PPHY | Plant Physiology | 植物生理学 |
| CS | Computer Science | 计算机 |
| BCHEM | Biochemistry | 生物化学 |
| OS | Operating Systems | 操作系统 |
| AMATH | Advanced Mathematics | 高等数学 |
| CNET | Computer Networks | 计算机网络 |
| LANG | Chinese Language | 语文 |
| GEO | Geography | 地理 |
| HIST | History | 历史 |
| IART | Introduction to Arts | 艺术概论 |
| ILING | Introduction to Linguistics | 语言学概论 |
| HSTUD | History Studies / Historiography | 历史学 |
| HFA | History of Foreign Art | 外国美术简史 |
| IARCH | Introduction to Archaeology | 考古学概论 |
| HACL | History of Ancient Chinese Literature | 中国古代文学史 |
| HWP | History of Western Philosophy | 西方哲学史 |
| POL | Politics | 政治 |
| IMOR | Ideology & Morality | 思想品德 |
| MGMT | Management | 管理学 |
| HRM | Human Resource Management | 人力资源管理 |
| TAX | Taxation | 税收学 |
| PSCI | Political Science | 政治学 |
| MARX | Marxist Theory | 马克思主义理论 |
| ELOG | Economic Logic | 经济学逻辑能力 |
| NJE | National Judicial Exam | 法考真题 |
| CLAW | Criminal Law | 刑法学 |
| CVLAW | Civil Law | 民法学 |
| LAW | Law / Jurisprudence | 法学 |
| TCM | Traditional Chinese Medicine | 中医综合 |
| WMED | Western Medicine | 西医综合 |
| NURS | Nursing | 护理学 |
| IT | Information Technology | 信息技术 |
| CEM | Civil Engineering Materials | 土木工程材料 |
| EDU | Education | 教育学 |
| PSY | Psychology | 心理学 |

Table 9: Bilingual names and abbreviations of all subject in the cultivation dimension

| Abbreviation | English Meaning | Chinese Name |
|---|---|---|
| *Major Categories* | | |
| TCS | Thinking & Cognitive Skills | 思维与认知能力 |
| EMH | Emotional & Mental Health | 情感与心理健康 |
| SIS | Social & Interpersonal Skills | 社会与人际交往 |
| CV | Character & Values | 品格与价值观 |
| PD | Personalized Development | 个性化发展 |
| TFS | Teaching Feedback & Support | 教学反馈与支持 |
| *Subcategories* | | |
| IC | Innovation & Creativity | 创新与创造力 |
| PSS | Problem-Solving Skills | 问题解决能力 |
| CT | Critical Thinking | 批判性思维 |
| GRL | Guided Reflective Learning | 反思性学习引导 |
| MA | Metacognitive Abilities | 元认知能力 |
| GKT | Guiding Knowledge Transfer | 引导知识迁移能力 |
| ER | Emotional Regulation | 情绪调控能力 |
| EC | Empathy & Compassion | 同理心与共情 |
| SCSE | Self-Confidence & Self-Efficacy | 自信心与自我效能感 |
| PR | Psychological Resilience | 心理韧性与抗挫力 |
| GM | Growth Mindset | 成长型思维 |
| TC | Teamwork & Collaboration | 团队协作能力 |
| ECOM | Effective Communication | 有效沟通能力 |
| SR | Social Responsibility | 社会责任感 |
| RA | Responsibility & Accountability | 责任感与担当 |
| IH | Integrity & Honesty | 正直与诚信 |
| PLP | Personalized Learning Paths | 个性化学习路径 |
| IDL | Interest-Driven Learning | 兴趣驱动学习 |
| HT | Heuristic Teaching | 启发式教学 |
| CTF | Constructive & Timely Feedback | 反馈的建设性与及时性 |

Table 10: Statistics of OmniEduBench for K-12 in the knowledge dimension.

| English Name | Chinese Name | 选择题 Multiple choice | 多选题 Multiple answer | 填空题 Fill-in-the-blank | 解答题 Short-answer | 复合题 Composite questions | 总计 Total |
|---|---|---|---|---|---|---|---|
| Chinese | 语文 | 350 | 8 | 1697 | 1261 | 51 | 3367 |
| Mathematics | 数学 | 527 | 12 | 1865 | 1181 | 142 | 3727 |
| Chemistry | 化学 | 274 | 76 | 799 | 477 | 14 | 1640 |
| History | 历史 | 67 | 24 | 63 | 211 | 5 | 370 |
| Geography | 地理 | 78 | 31 | 277 | 173 | 4 | 563 |
| Moral Education | 思想品德 | 14 | 30 | 34 | 56 | 4 | 138 |
| Politics | 政治 | 260 | 241 | 281 | 64 | 12 | 858 |
| Physics | 物理 | 82 | 15 | 178 | 46 | 16 | 337 |
| Biology | 生物 | 115 | 94 | 360 | 124 | 0 | 693 |
| Nature Science | 自然与科学 | 8 | 0 | 23 | 22 | 0 | 53 |
| Information Technology | 信息技术 | 18 | 2 | 14 | 1 | 1 | 36 |
| Total | 总计 | 1793 | 533 | 5591 | 3616 | 249 | 11.782K |

Table 11: Statistics of OmniEduBench for high, college, and professional schools in the knowledge dimension

| English Name | Chinese Name | Single Choice 单选题 | Multiple choice 多选题 | Term explanation 名词解释 | Short answer 简答题 | Essay 论述题 | Case analysis 案例分析题 | Fill-in-blank 填空题 | Calculation 计算题 | True/False 判断题 | Logical reasoning 逻辑推理 | Total 总计 |
|---|---|---|---|---|---|---|---|---|---|---|---|---|
| Traditional Chinese Medicine | 中医综合 | 230 | 317 | 0 | 0 | 0 | 0 | 0 | 0 | 0 | 0 | 547 |
| Chinese Ancient Literary History | 中国古代文学史 | 31 | 14 | 20 | 27 | 19 | 0 | 0 | 0 | 0 | 0 | 111 |
| Human Resource Management | 人力资源管理 | 36 | 35 | 9 | 38 | 19 | 6 | 4 | 0 | 0 | 0 | 147 |
| Jurisprudence | 法学 | 109 | 153 | 0 | 0 | 0 | 0 | 0 | 0 | 0 | 0 | 262 |
| Criminal Law | 刑法学 | 158 | 171 | 2 | 3 | 1 | 0 | 0 | 0 | 0 | 0 | 335 |
| History | 历史学 | 26 | 0 | 68 | 6 | 17 | 12 | 0 | 0 | 0 | 0 | 129 |
| Civil Engineering Materials | 土木工程材料 | 36 | 2 | 67 | 21 | 57 | 2 | 107 | 39 | 3 | 0 | 334 |
| A Brief History of Foreign Art | 外国美术简史 | 0 | 0 | 61 | 37 | 15 | 14 | 0 | 0 | 0 | 0 | 127 |
| Psychology | 心理学 | 146 | 45 | 0 | 35 | 0 | 1 | 0 | 0 | 0 | 0 | 227 |
| Nursing | 护理学 | 61 | 41 | 0 | 5 | 0 | 16 | 0 | 0 | 0 | 0 | 123 |
| Operating Systems | 操作系统 | 98 | 0 | 0 | 0 | 0 | 59 | 0 | 0 | 0 | 0 | 157 |
| Politics | 政治 | 7 | 47 | 0 | 0 | 0 | 36 | 0 | 0 | 0 | 0 | 90 |
| Pedagogy | 教育学 | 179 | 0 | 0 | 45 | 39 | 0 | 0 | 0 | 14 | 0 | 277 |
| Advanced Mathematics | 高等数学 | 43 | 0 | 0 | 0 | 3 | 0 | 40 | 55 | 0 | 0 | 141 |
| Plant Physiology | 植物生理学 | 66 | 0 | 0 | 134 | 33 | 48 | 0 | 0 | 0 | 0 | 281 |
| Probability and Mathematical Statistics | 概率论与数理统计 | 0 | 0 | 0 | 2 | 0 | 74 | 0 | 257 | 0 | 0 | 333 |
| Civil Law | 民法学 | 91 | 187 | 0 | 0 | 0 | 0 | 0 | 0 | 0 | 0 | 278 |
| judicial Practice | 法考真题 | 193 | 430 | 0 | 0 | 0 | 0 | 0 | 0 | 0 | 0 | 623 |
| Biochemistry | 高等生物化学 | 62 | 0 | 0 | 70 | 23 | 0 | 0 | 0 | 0 | 0 | 155 |
| Taxation | 税收学 | 19 | 2 | 7 | 30 | 0 | 2 | 0 | 23 | 11 | 0 | 94 |
| Management | 管理学 | 0 | 0 | 0 | 0 | 0 | 0 | 0 | 0 | 0 | 202 | 202 |
| Economics | 经济学逻辑能力 | 2 | 0 | 0 | 0 | 0 | 0 | 0 | 0 | 0 | 58 | 60 |
| Archaeology | 考古学概论 | 0 | 0 | 121 | 0 | 0 | 0 | 0 | 0 | 0 | 0 | 121 |
| Introduction to Archaeology | 艺术学概论 | 57 | 43 | 80 | 45 | 26 | 0 | 0 | 0 | 37 | 0 | 288 |
| Western Medicine | 西医综合 | 159 | 119 | 0 | 0 | 0 | 0 | 0 | 0 | 0 | 0 | 278 |
| Western Philosophy | 西方哲学史 | 51 | 0 | 0 | 0 | 0 | 0 | 0 | 0 | 0 | 0 | 51 |
| Computer Science | 计算机 | 137 | 0 | 0 | 0 | 0 | 47 | 0 | 0 | 0 | 0 | 184 |
| Computer Networks | 计算机网络 | 91 | 0 | 0 | 0 | 0 | 50 | 0 | 0 | 0 | 0 | 141 |
| Introduction to Linguistics | 语言学概论 | 53 | 29 | 24 | 22 | 26 | 18 | 0 | 0 | 0 | 0 | 172 |
| Marxist Theory | 马克思主义理论 | 0 | 0 | 20 | 31 | 13 | 7 | 0 | 0 | 0 | 0 | 71 |
| Total 总计 | | 2141 | 1635 | 479 | 551 | 291 | 392 | 151 | 374 | 65 | 260 | 6339 |

Table 12: Statistics of OmniEduBench for 20 subjects in the cultivation dimension.

| English Name | Chinese Name | Count |
|---|---|---|
| Emotional Regulation Skills | 情绪调控能力 | 325 |
| Innovation & Creativity | 创新与创造力 | 275 |
| Heuristic Teaching | 启发式教学 | 434 |
| Sense of Responsibility & Accountability | 责任感与担当 | 330 |
| Problem-Solving Skills | 问题解决能力 | 288 |
| Team Collaboration Skills | 团队协作能力 | 291 |
| Empathy & Sympathy | 同理心与共情 | 385 |
| Self-Confidence & Self-Efficacy | 自信心与自我效能感 | 358 |
| Constructiveness & Timeliness of Feedback | 反馈的建设性与及时性 | 196 |
| Integrity & Honesty | 正直与诚信 | 371 |
| Psychological Resilience & Anti-Frustration Ability | 心理韧性与抗挫力 | 393 |
| Personalized Learning Paths | 个性化学习路径 | 292 |
| Reflective Learning | 反思性学习引导 | 224 |
| Guiding Knowledge Transfer | 知识迁移能力 | 384 |
| Metacognitive Skills | 元认知能力 | 338 |
| Interest-Driven Learning | 兴趣驱动学习 | 320 |
| Critical Thinking | 批判性思维 | 428 |
| Growth Mindset | 成长型思维 | 393 |
| Social Responsibility | 社会责任感 | 317 |
| Effective Communication Skills | 有效沟通能力 | 139 |
| Total | 总计 | 6.481K |

我班上有学生因家庭暴力表现出攻击性行为，其他老师建议开除他。我认为他需要帮助而非惩罚。如何说服同事？请选择最能体现'同理心与共情'和'问题解决能力'的回答：
**A.** 这个孩子本质不坏，他只是用攻击来保护自己。我们应该分析他的行为背后的原因，制定支持计划。
**B.** 开除他只会让他更恨社会，不如我们联合心理老师做一次行为评估，再决定干预方案。
**C.** 反正他又不是我亲生的，你决定吧，别到时候出事怪我没提醒。
**D.** 我觉得他应该去特殊学校，普通班根本管不了这种孩子。

A student in my class exhibited aggressive behavior due to domestic violence, and other teachers recommended expelling him. I think he needs help rather than punishment. How to convince a colleague? Please choose the answer that best reflects 'empathy and empathy' and 'problem-solving skills': A. This child is not bad by nature, he just uses aggression to protect himself. We should analyze the reasons behind his behavior and develop a support plan.
B. Expelling him will only make him hate society more, so why not do a behavioral assessment with a psychologist and then decide on an intervention plan.
C. Anyway, he is not my biological son, you decide, don't blame me for not reminding me when something happens.
D. I think he should go to a special school, ordinary classes can't control this kind of child at all.

答案：B

Answer: B

Question within the cultivation dimension

我在背诵古诗词时总觉得枯燥，记不住，也不知道有什么用。怎么才能让传统文化学习变得有意义？请选择最能体现'兴趣驱动学习'和'知识迁移能力'的回答：
**A.** 你得多抄几遍，熟能生巧。
**B.** 考试要考，必须死记硬背下来。
**C.** 试着把这些诗句用在日记或作文里，比如用'长风破浪会有时'鼓励自己面对考试压力，你会发现它有现实力量。
**D.** 找一首你喜欢的流行歌，看看有没有引用古诗的歌词，比较一下意境。

I always feel bored when reciting ancient poems, I can't remember them, and I don't know what they are used for. How can we make traditional culture learning meaningful? Please choose the answer that best reflects "interest-driven learning" and "knowledge transfer ability":
A. You have to copy it several times, practice makes perfect.
B. To take the exam, you must memorize it.
C. Try to use these verses in your diary or composition, such as 'There will be times when the wind and waves will be long' to encourage yourself to face the pressure of the exam, and you will find that it has practical power.
D. Find a pop song you like, see if there are lyrics quoted from ancient poems, and compare the artistic conception.
.

答案：C

Answer: C

Question within the cultivation dimension

Figure 8: Examples of different questions in the knowledge dimension and cultivation dimensions.

已知抛物线Γ：y² = 4x的焦点为F，过F作斜率为k的直线交Γ于A、B两点。
（1）若|AB| = 10，求线段AB中点的横坐标；（2）证明：在抛物线Γ上所有的点中，顶点到焦点F的距离最小；（3）证明：以AB为直径的圆与抛物线Γ的准线相切。

It is known that the focus of the parabolic Γ: y²=4x is F, and the straight line with a slope of k crosses F Γ at points A and B. (1) If | AB |=10, find the abscissa of the midpoint of the line segment AB; (2) Prove: Among all the points on the parabolic Γ, the distance from the vertex to the focal point F is the smallest; (3) Proof: The circle with AB as the diameter is tangent to the collimator of the parabolic Γ.

答案： （1）4；（2）设抛物线上任意一点P(x, y)，其到焦点F(1, 0)的距离为|PF|=√((x-1)²+y²)=√((x-1)²+4x)=√(x²+2x+1)=|x+1|。因x≥0，故|PF|=x+1。当x=0时，|PF|取最小值1，此时P为顶点(0, 0)。故顶点到焦点F的距离最小。（3）设A(x₁, y₁)，B(x₂, y₂)，AB中点为M。由抛物线定义，
|AB|=|AF|+|BF|=(x₁+1)+(x₂+1)=x₁+x₂+2。以AB为直径的圆，其圆心M的横坐标为(x₁+x₂)/2，半径R=|AB|/2=(x₁+x₂+2)/2。圆心M到准线x=-1的距离d=|(x₁+x₂)/2 - (-1)|=(x₁+x₂+2)/2。因d=R，故该圆与准线相切

Answer： （1）4; (2) Suppose any point P(x, y) on the parabola has a distance from the focus Γ(1, 0) as | PF |=√((x-1)²+y²)=√((x-1)²+4x)=√(x²+2x+1)=|x+1|。 Because x≥0, so | PF |=x+1。 When x=0, | PF | take the minimum value of 1, where P is the vertex (0, 0). Therefore, the distance from the vertex to the focal point F is the smallest. (3) Let A(x₁, y₁), B(x₂, y₂), and the midpoint of AB is M. Defined by parabola, | AB |=| AF |+| BF |=(x₁+1)+(x₂+1)=x₁+x₂+2。 For a circle with AB as the diameter, the abscissa of the center M is (x₁+x₂)/2, and the radius R=| AB |/2=(x₁+x₂+2)/2。 The distance from the center of the circle M to the collimator x=-1 d=| (x₁+x₂)/2 - (-1)|=(x₁+x₂+2)/2。 Since d = R, the circle is tangent to the collimator

Figure 9: Examples of different questions in the knowledge dimension and cultivation dimensions.

