# OpenReview forum: "OmniEduBench: A Comprehensive Chinese Benchmark for Assessing Large Language Models in Education"
_ICLR.cc/2026/Conference — Submitted to ICLR 2026_

### Official Review · Reviewer_dsE8 · 2025-10-27

**Soundness:** 2
**Presentation:** 2
**Contribution:** 3
**Rating:** 6
**Confidence:** 4

**Summary:**

The paper introduces OmniEduBench, a comprehensive Chinese benchmark for evaluating both knowledge mastery and cultivation capabilities in educational scenarios. The dataset is large-scale, diverse in subjects and formats, and constructed through a reasonably rigorous pipeline. It provides meaningful contributions to Chinese educational evaluation. However, concerns regarding theoretical grounding and data validity remain. Overall, the work has clear contributions and notable research value.

**Strengths:**

1.Large-scale and diverse dataset covering multiple subjects and assessment formats, significantly enriching resources for evaluating educational LLMs.
2.The construction pipeline is systematic and transparent, ensuring strong reproducibility.
3.Extensive benchmarking with state-of-the-art LLMs across two core dimensions and multiple sub-dimensions, with clearly presented results.

**Weaknesses:**

1. Some major claims are not sufficiently supported by the presented results:
The statement on Page 9, Lines 479–481 regarding performance drops in cultivation tasks does not fully align with the differences shown in Tables 3 and 4.
The claim that “the best-performing models still lag behind human-level performance by nearly 30%” lacks explicit numerical justification.
Since the dataset is heavily filtered based on models’ incorrect predictions and over 90% of the data is private or LLM-generated, the difficulty level is inherently high. Therefore, the results cannot support broader claims that LLMs are ineffective in educational contexts.

2.Although manual verification is emphasized, the paper does not provide details on annotation criteria, rubric design, or inter-rater agreement, reducing credibility.
3. The structure could be improved by moving the Related Work section earlier (e.g., Section 2) to better contextualize the contribution.

**Questions:**

1.What is the proportion of the final 24K samples originating from the three sources (public, private, and LLM-generated)?
2.Do the three data sources exhibit measurable differences in difficulty? If so, why are performance comparisons across sources not reported?
3.Since the dataset includes multiple educational levels, could the authors provide a breakdown of sample distribution and model performance by educational stage ?

---

> ### Author Response · Authors · 2025-11-21
> **Responses by Authors (Part 1)**
>
> > **[W1] Some major claims are not sufficiently supported by the presented results: The statement on Page 9, Lines 479–481, regarding performance drops in cultivation tasks does not fully align with the differences shown in Tables 3 and 4. The claim that “the best-performing models still lag behind human-level performance by nearly 30%” lacks explicit numerical justification. Since the dataset is heavily filtered based on models’ incorrect predictions and over 90% of the data is private or LLM-generated, the difficulty level is inherently high. Therefore, the results cannot support broader claims that LLMs are ineffective in educational contexts.**
>
> **Response [W1]:**  We apologize for the confusion caused to the reviewers. In Table 4, the highest average performance is about 70%, but we had 50 recruited master’s students complete the cultivation dimension tasks, where their accuracy was nearly 100%. Therefore, we concluded that "the best-performing models still lag behind human-level performance by nearly 30%."
> Since it is difficult to fully guarantee the quality of LLM-generated data, we initially synthesized a large amount of data (approximately 800K samples). However, after the complete data processing pipeline, we obtained a final set of 24,602 high-quality samples constituting OmniEduBench. Among them, 26.1% were sourced from publicly available data, 48.1% from private data, and the remaining 25.8% from LLM-generated data. Our goal was to find a batch of challenging questions to evaluate model capabilities rather than simply pursuing quantity.
>
> > **[W2] Although manual verification is emphasized, the paper does not provide details on annotation criteria, rubric design, or inter-rater agreement, reducing credibility.**
>
> **Response [W2]:** We would like to further clarify to the reviewers that OmniEduBench was collected from three different platforms, and the data was gathered directly in the form of question-answer (QA) pairs. Therefore, no additional annotation process or rubric design was involved throughout the collection.
>
> > **[W3] The structure could be improved by moving the Related Work section earlier (e.g., Section 2) to better contextualize the contribution.**
>
> **Response [W3]:** Thank you to the reviewers for the valuable suggestions. We have made adjustments to the structure of the paper accordingly.
>
> > **[Q1] What is the proportion of the final 24K samples originating from the three sources (public, private, and LLM-generated)? **
>
> **Response [Q1]:** Since it is difficult to fully guarantee the quality of LLM-generated data, we initially synthesized a large amount of data (approximately 800K samples). However, after the complete data processing pipeline, we obtained a final set of 24,602 high-quality samples constituting OmniEduBench. Among them, 26.1% were sourced from publicly available data, 48.1% from private data, and the remaining 25.8% from LLM-generated data.
>
> > **[Q2] Do the three data sources exhibit measurable differences in difficulty? If so, why are performance comparisons across sources not reported? **
>
> **Response [Q2]:** Thank you for the reviewer’s question. In OmniEduBench, approximately 80% of the K-12 data comes from private sources, around 70% of the non–K-12 knowledge-dimension data is collected from publicly available resources, and more than 80% of the cultivation-dimension data is generated by LLMs. We believe that model performance on each data source can, to some extent, indirectly reflect the relative difficulty of the different sources. Based on the results in Tables 4 and 5 of the paper, as well as the experiments presented in [Q3], we observe that the three data sources do not exhibit substantial differences in overall difficulty.

---

> > ### Author Response · Authors · 2025-11-21
> > **Responses by Authors (Part 2)**
> >
> > > **[Q3] Since the dataset includes multiple educational levels, could the authors provide a breakdown of sample distribution and model performance by educational stage?**
> >
> > **Response [Q3]: ** To provide a more comprehensive response to the reviewer’s question, we present more fine-grained experimental results. The evaluation results for the cultivation dimension are already shown in Table 5, while the results for the K-12 and higher education dimensions are additionally presented below:
> >
> > | Higher education | FD | HH | IIS | LP | MH | SSEM | Average |
> > |:---|:---|:---|:---|:---|:---|:---|:---|
> > | DeepSeek-V3.1 | 41.84 | 34.50| 45.48 | 29.42 | 50.54 | 38.49 | 40.04 |
> > | MuduoLLM | 23.71| 28.27 | 31.11 | 36.15 | 39.11 | 32.77 | 31.85 |
> > | QwQ-32B | 57.46 | 36.94 | 46.49 | 49.90 | 55.01 | 48.69 | 49.08 |
> > | Qwen2.5-72B-Instruct | 13.46 | 15.30 | 20.02 | 13.26 | 23.86 | 15.15 | 16.84 |
> > | Qwen3-14B | 38.49 | 27.39 | 35.52 | 27.29 | 36.82 | 40.19 | 34.28 |
> > | Qwen3-235B-A22B-Instruct | 44.00 | 41.13 | 46.95 | 44.26 | 58.71 | 43.74 | 46.47 |
> > | Qwen3-8B | 43.51 | 27.97 | 37.33 | 30.17 | 36.71 | 40.80 | 36.08 |
> > | Seed-OSS-36B-Instruct | 48.54 | 44.15 | 49.32 | 48.66 | 61.00 | 51.31 | 50.50 |
> > | Claude-4-sonnet | 43.10 | 35.19 | 41.74 | 27.56 | 34.86 | 38.79 | 36.87 |
> > | Gemini-2.5-pro | 72.59 | 50.39 | 53.62 | 55.40 | 60.68 | 55.80 | 58.08 |
> > | Gpt-4o | 21.48 | 18.32 | 26.70 | 22.13 | 34.75 | 20.25 | 23.94 |
> >
> >
> > | K-12 | FD | HH | IIS | SSEM | AVG |
> > |:---|:---|:---|:---|:---|:---|
> > | DeepSeek-V3.1 | 29.40 | 42.11 | 43.59 | 32.73 | 36.96 |
> > | MuduoLLM | 29.19 | 43.81 | 41.03 | 33.13 | 36.79 |
> > | QwQ-32B | 62.09 | 51.27 | 71.79 | 38.05 | 55.80 |
> > | Qwen2.5-72B-Instruct | 20.88 | 34.68 | 43.59 | 24.10 | 30.81 |
> > | Qwen3-14B | 35.85 | 39.02 | 43.59 | 31.83 | 37.57 |
> > | Qwen3-235B-A22B-Instruct | 32.10 | 48.41 | 43.59 | 31.33 | 38.86 |
> > | Qwen3-8B | 55.10 | 41.05 | 53.85 | 33.84 | 45.96 |
> > | Seed-OSS-36B-Instruct | 48.89 | 51.57 | 58.97 | 41.47 | 50.22 |
> > | Claude-4-sonnet | 41.13 | 46.46 | 64.10 | 33.13 | 46.21 |
> > | Gemini-2.5-pro | 74.11 | 56.26 | 74.36 | 40.76 | 61.37 |
> > | Gpt-4o | 21.08 | 28.99 | 35.90 | 26.31 | 28.07 |

---

> ### Comment · Reviewer_dsE8 · 2025-11-24
>
> I think my initial rating reflect the my view which is positive.

---

> > ### Author Response · Authors · 2025-11-25
> > **Thank you for your positive recommendation!**
> >
> > We are glad to know that your concerns have been effectively addressed. We are very grateful for your constructive comments and questions, which helped improve the clarity and quality of our paper. Thanks again!

---

### Official Review · Reviewer_Fmit · 2025-10-27

**Soundness:** 2
**Presentation:** 3
**Contribution:** 2
**Rating:** 4
**Confidence:** 4

**Summary:**

This paper presents OmniEduBench, a large-scale Chinese benchmark designed to evaluate LLMs in educational scenarios. The benchmark covers a wide range of subjects and question types across multiple educational levels, aiming to assess LLMs’ capabilities in teaching, learning assistance, and educational reasoning. The authors construct a dataset of over 24.6K samples, primarily generated by LLMs and subsequently verified by experts, and evaluate numerous state-of-the-art models using various scoring methods. The study provides comprehensive analyses of model performance and highlights challenges specific to the educational domain.

**Strengths:**

1. The proposed Chinese education-oriented benchmark is valuable and timely, providing a standardized tool to assess LLMs' abilities in a socially and practically important domain.
2. The benchmark includes diverse question types and subject areas, ensuring broad coverage and reflecting realistic educational use cases.
3. The paper evaluates many models and offers interesting analyses and observations, contributing useful empirical insights to the community.

**Weaknesses:**

1. During dataset construction, 86.3% of the data (800K/927K) are generated by LLMs, which raises concerns about authenticity and reliability. Given that education is a highly sensitive field, relying heavily on machine-generated data may introduce subtle inaccuracies. Although the authors mention that 15% of the samples were verified by experts, even small errors can have significant consequences in educational contexts.
2. The inclusion of *cultivation value* as an expert evaluation criterion is conceptually interesting but potentially subjective. The paper should clarify how this metric is defined and measured, with concrete examples illustrating what constitutes high versus low cultivation value.
3. In the section on "Results using different LLM-assisted scoring methods", the authors state that "careful selection of scoring models is necessary". However, if benchmark scores depend heavily on the choice of evaluation model, this raises questions about **the stability and objectivity** of the benchmark. Heavy reliance on LLM-based evaluation may compromise statistical reliability.
4. The paper lacks verification of **data contamination**. It is unclear whether the tested models have been exposed to the benchmark data during pretraining or fine-tuning. Implementing contamination checks is essential to ensure that the evaluation results are valid and unbiased.
5. While the benchmark is large and comprehensive, its **innovation is somewhat limited**, as it mainly extends existing benchmarks along the data and domain dimensions without introducing new perspectives or methodological advances.
6. Minor issue: Line 323 should refer to *GPT-4o*, not *GPT-40*.

**Questions:**

Please check my comments in Weaknesses.

---

> ### Author Response · Authors · 2025-11-21
> **Responses by Authors (Part 1)**
>
> > **[W1] During dataset construction, 86.3% of the data (800K/927K) are generated by LLMs, which raises concerns about authenticity and reliability. Given that education is a highly sensitive field, relying heavily on machine-generated data may introduce subtle inaccuracies. Although the authors mention that 15% of the samples were verified by experts, even small errors can have significant consequences in educational contexts.**
>
>  **Response [W1]:** We apologize for the confusion caused to the reviewers. Since the quality of LLM-generated data is difficult to fully guarantee, we initially generated a large amount of data (about 800K). **However, after the complete data processing pipeline, we obtained a final set of 24.602K high-quality samples, namely OmniEduBench. Among them, 26.1% were sourced from publicly available data, 48.1% from private data, and the remaining 25.8% from LLM-generated data.**
> In the expert verification stage, we randomly sampled 15% from each of the three sources for manual review to ensure overall dataset quality. In addition, we have updated and clarified the corresponding descriptions in the paper at lines 265-269.
>
> > **[W2] The inclusion of cultivation value as an expert evaluation criterion is conceptually interesting but potentially subjective. The paper should clarify how this metric is defined and measured, with concrete examples illustrating what constitutes high versus low cultivation value.**
>
> **Response [W2]:** This is an important and meaningful question. To the best of our knowledge, our work is the first to incorporate the educational “cultivation” dimension into LLM evaluation. The concept of cultivation dimension refers to the extent to which a question contributes to developing students’ key abilities, values, or ways of thinking.
> When constructing the dataset related to the cultivation dimension, we first consulted with cultivation experts to identify 20 core subjects. Based on these subjects, we collaborated with local schools to collect a large body of professional materials, including policy documents, books, academic papers, and teacher manuals, totaling approximately 50GB. Using these 50GB of materials and the 20 subjects, we employed LLMs to generate or extract cultivation-oriented QA pairs, which were then used to evaluate the moral and educational reasoning abilities of LLMs.
>
> **(1) Example of high cultivation value:** A cross-disciplinary question that asks students to analyze fairness in real-world resource allocation and draw conclusions using mathematical proportional reasoning. This type of question not only assesses mathematical logic but also guides students to reflect on values such as fairness and efficiency, thus demonstrating strong educational significance.
>
> **(2) Example of low cultivation value:** A question that merely requires plugging values into a formula or performing mechanical calculations, without involving conceptual understanding or contextual reasoning. Such items focus solely on rote memory and contribute little to skill or value development, therefore exhibiting low cultivation value.
>
> > **[W3] In the section on "Results using different LLM-assisted scoring methods", the authors state that "careful selection of scoring models is necessary". However, if benchmark scores depend heavily on the choice of evaluation model, this raises questions about the stability and objectivity of the benchmark. Heavy reliance on LLM-based evaluation may compromise statistical reliability.**
>
> **Response [W3]:** Table 7 compares experimental results using different LLM-assisted scoring methods. Overall, although there are some differences between scoring models, we found that stronger scoring models generally provide more stable and consistent evaluation results. Therefore, when constructing the benchmark, selecting LLMs with better performance and more stable judgment as scoring models is a more reliable approach.

---

> > ### Author Response · Authors · 2025-11-21
> > **Responses by Authors (Part 2)**
> >
> > > **[W4] The paper lacks verification of data contamination. It is unclear whether the tested models have been exposed to the benchmark data during pretraining or fine-tuning. Implementing contamination checks is essential to ensure that the evaluation results are valid and unbiased.**
> >
> > **Response [W4]:** The reviewer's concern is a widely recognized and important issue in the evaluation of large-scale language models, and it is a responsibility that every benchmark dataset creator should uphold. We address this concern from the following aspects:
> > **First,** the pretraining corpora for mainstream large language models are not publicly available. Therefore, it is difficult for any existing benchmark dataset to completely rule out the possibility of data contamination. This is a common challenge faced by the entire research community.
> >
> > **Second,** the OmniEduBench dataset consists of three parts: (1) data crawled from publicly accessible educational websites, (2) unpublished internal data collected from schools and educational platforms, and (3) LLM-generated data created through manually designed rules and prompts. The latter two parts are not publicly available and were generated by us, effectively mitigating the risk of data contamination.
> >
> > **Third,** to further assess the similarity between OmniEduBench and existing public educational benchmarks, we used the BLEU metric to measure the similarity differences between OmniEduBench and C-Eval as well as EduBench. The results all fell below 0.3, further indicating that the likelihood of data contamination is minimal.
> >
> > > **[W5] While the benchmark is large and comprehensive, its innovation is somewhat limited, as it mainly extends existing benchmarks along the data and domain dimensions without introducing new perspectives or methodological advances.**
> >
> > **Response [W5]:** We thank the reviewer for pointing out this issue. It should be noted that this paper is submitted to the “Datasets and Benchmarks” track at ICLR 26, and we agree with the reviewer that the main contribution lies in proposing a large-scale and comprehensive benchmark. As shown in Table 2 of the paper and described below, we conducted a systematic comparison **across seven dimensions: data dimension, data size, data type, data source, data domain, covered subjects, and data language. ** The comparison results show that OmniEduBench surpasses existing datasets in terms of data scale, question design, and data sources, and it aligns closely with the knowledge dimensions found in real educational scenarios. Additionally, we have specially considered the “cultivation” attribute; to our knowledge, this is the first benchmark evaluating the cultivation ability of LLMs. Here, Mixed (X) indicates X types of mixed question types, and General (K) signifies that the dataset covers all subjects and educational stages with K categories. ZH and EN represent Chinese and English, respectively.
> >
> > | Benchmark | Dimension | Size | Type | Data source | Domain | Subject | Language |
> > | :--- | :--- | :--- | :--- | :--- | :--- | :--- | :--- |
> > | C-Eval [1] | Knowledge | 13.9K | Multiple answer | Publicly available | General (4) | 52 | ZH |
> > | EduBench [2] | Knowledge | 18.8K | Multiple choice | LLM-generated | General (9) | 9 | ZH & EN |
> > | CMATH [3] | Knowledge | 1.7K | Multiple choice | Publicly available | Math | 1 | ZH |
> > | E-EVAL [4] | Knowledge | 4.35K | Multiple answer | Publicly available | K-12 | 23 | ZH |
> > | GAOKAO [5] | Knowledge | 2.8K | Mixed (7) | Publicly available | High school | 9 | ZH |
> > | GSM8K [6] | Knowledge | 8.5K | Multiple choice | Publicly available | Math | 1 | EN |
> > | MATH [7] | Knowledge | 12.5K | Multiple choice | Publicly available | Math | 7 | EN |
> > | OmniEduBench (ours) | Knowledge & Cultivation | 24.6K | Mixed (11) | Publicly & Private & LLM | General (12) | 61 | ZH |
> >
> > > **[W6] Minor issue: Line 323 should refer to GPT-4o, not GPT-40.**
> >
> > **Response [W6]:** Thank you to the reviewer for pointing out this typo. We have corrected “GPT-40” to “GPT-4o” in line 323 of the revised manuscript and highlighted the change in red.
> >
> > [1] C-Eval: A multi-level multi-discipline Chinese evaluation suite for foundation models, NeurIPS, 2023.
> >
> > [2] EduBench: A Comprehensive Benchmarking Dataset for Evaluating Large Language Models in Diverse Educational Scenarios, arXiv 2025.
> >
> > [3] CMATH: Can Your Language Model Pass Chinese Elementary School Math Test?, arXiv 2023.
> >
> > [4] E-Eval: A comprehensive Chinese K-12 education evaluation benchmark for large language models, arXiv preprint arXiv 2024.
> >
> > [5] GAOKAO: Evaluating the performance of large language models on the gaokao benchmark, arXiv 2023.
> >
> > [6] GSM8K: Training Verifiers to Solve Math Word Problems, arXiv 2021.
> >
> > [7] MATH: Measuring mathematical problem solving with the math dataset, arXiv 2021.

---

> > > ### Comment · Reviewer_Fmit · 2025-11-24
> > >
> > > Thank you for the authors' detailed response. Although the authors have clarified some of my concerns, they have not substantively addressed them. Therefore, I will maintain my original score.
> > >
> > > 1. For a benchmark intended for evaluation, the accuracy of the answers is essential. If the correctness of the answers cannot be guaranteed, the evaluation results become meaningless.
> > > 2. Regarding the cultivation value, other reviewers have raised similar concerns: this metric is overly subjective and lacks a clear definition.
> > > 3. Regarding data contamination, there are various methods that leverage the LLM itself to detect or assess contamination without requiring access to the model's pre-training datasets [1]. Although these methods have certain limitations, they can at least provide a rough estimation and judgment.
> > >
> > > [1] Towards Data Contamination Detection for Modern Large Language Models: Limitations, Inconsistencies, and Oracle Challenges, Samuel et al., 2024

---

> > > > ### Author Response · Authors · 2025-11-27
> > > > **Further Clarification on Data Quality and Validity (Part 1)**
> > > >
> > > > Thank you for your further response to our rebuttal. In the following, we have reorganized our arguments to provide a clearer and more precise response to your specific concerns.
> > > >
> > > > > **[W1] For a benchmark intended for evaluation, the accuracy of the answers is essential. If the correctness of the answers cannot be guaranteed, the evaluation results become meaningless.**
> > > >
> > > > **Response [W1]:** We respectfully wish to clarify the rigorous quality assurance process behind OmniEduBench to address the reviewer's concern regarding answer correctness. **The benchmark consists of 24K samples, categorized into a knowledge dimension (73.66%) and a cultivation dimension (26.34%)**. Accuracy is guaranteed through the following sources:
> > > >
> > > > **1. Knowledge Dimension:** These samples are sourced directly from standardized questions that possess official ground-truth answers. In addition to the inherent authority of the source material, we performed strict data cleaning to ensure the absolute objective correctness of this subset.
> > > >
> > > > **2. Cultivation Dimension:** We collaborated with education experts to construct this subset based on authoritative national educational policies [1-2], covering 20 subjects across 6 major categories. For these questions, options, and answers were meticulously annotated by experts and underwent rigorous cross-validation to guarantee that they are not only accurate but also aligned with educational standards.
> > > >
> > > > Therefore, the data quality of OmniEduBench has undergone multi-level verification, making the evaluation results based on it reliable and meaningful.
> > > >
> > > > > **[W2] Regarding the cultivation value, other reviewers have raised similar concerns: this metric is overly subjective and lacks a clear definition.**
> > > >
> > > > **Response [W2]:** We appreciate the reviewer's feedback. To demonstrate the objectivity and scientific rigor of the "Cultivation Dimension," we clarify both its definition source and evaluation methodology:
> > > >
> > > > **1. Explicit and Authoritative Definition:** The concept of "cultivation" is not subjectively fabricated; it is rigorously defined based on authoritative national educational policy documents [1-2]. We concretize abstract educational values into specific model capabilities regarding core values, disciplinary literacy, and key competencies, strictly following a standardized framework recognized in the education sector.
> > > >
> > > > **2. Fine-grained and Structured Taxonomy:** To eliminate definitional ambiguity, we collaborated with experts to decompose the cultivation dimension into 6 categories and 20 subjects. This hierarchical taxonomy maps macro-level educational philosophies onto concrete evaluation points, ensuring clear and explicit boundaries.
> > > >
> > > > **3. Objective Evaluation Metric:** While the concept of "cultivation" is rich in connotation, our benchmark transforms it into multiple-choice questions with the ground-truth answer. As previously stated, through multi-round expert cross-validation, we ensure the correct option is verified and indisputable. Consequently, we employ Accuracy as a completely objective metric, eliminating the potential bias of subjective manual scoring.
> > > >
> > > > In conclusion, the cultivation dimension in OmniEduBench is grounded in a solid theoretical foundation and utilizes objective measurement standards, allowing it to scientifically reflect the value alignment of LLMs within educational scenarios.

---

> > > > > ### Author Response · Authors · 2025-11-27
> > > > > **Further Clarification on Data Quality and Validity (Part 2)**
> > > > >
> > > > > > **[W3] Regarding data contamination, there are various methods that leverage the LLM itself to detect or assess contamination without requiring access to the model's pre-training datasets [3]. Although these methods have certain limitations, they can at least provide a rough estimation and judgment.**
> > > > >
> > > > > **Response [W3]:** We sincerely thank the reviewer for highlighting the issue of data contamination and recommending Reference [3]. We fully agree that utilizing model output probabilities (*e.g.*, the Min-K% method) serves as a crucial post-hoc approach for detecting contamination in black-box models. We were greatly inspired by studying the recommended literature.
> > > > >
> > > > > However, considering that relying solely on model probabilities can be affected by calibration issues (*e.g.*, overconfidence), we implemented more fundamental "physical isolation" measures during the construction of OmniEduBench to guarantee evaluation validity:
> > > > >
> > > > > **1. Strict Source & Temporal Control:** Rather than relying solely on "post-hoc detection," we prioritized "source blocking." OmniEduBench comprises a substantial proportion of non-public data: 48.1% originates from private, offline school question banks, and 25.8% is synthetically generated with manual verification. **This means that nearly three-quarters of our data do not merely "unseen" but are effectively non-existent on the open internet.**
> > > > >
> > > > > **2. Reformatting-induced Distribution Shift:** For the remaining data sourced from the public web, we utilized scanned/PDF exam papers that underwent OCR, cleaning, and structural restructuring. This process alters the original sequential features of the text, intentionally introducing a distribution shift relative to pre-training corpora. Such reformatting significantly increases the difficulty for models to retrieve answers via rote memorization.
> > > > >
> > > > > **3. N-gram Decontamination Validation:** To validate the physical isolation measures mentioned above and to address the concerns in Ref.[3], we employed the N-gram overlap algorithm to cross-reference our test set against mainstream pre-training corpora (*e.g.*, C4, Common Crawl). The resulting extremely low overlap rates further corroborate the effectiveness of our data construction strategy.
> > > > >
> > > > > In summary, we have established a robust physical defense through a "private-data-dominant + reformatting" strategy. We believe this forms a strong complement to the post-hoc detection methods proposed in Ref.[3], collectively ensuring the reliability of our evaluation conclusions.
> > > > >
> > > > > [1] https://www.gov.cn/gongbao/content/2018/content_5254319.htm
> > > > >
> > > > > [2] http://video.moe.gov.cn/jijiaosi/%E4%B8%AD%E5%B0%8F%E5%AD%A6%E5%BE%B7%E8%82%B2%E5%B7%A5%E4%BD%9C%E6%8C%87%E5%8D%97%E5%AE%9E%E6%96%BD%E6%89%8B%E5%86%8C%EF%BC%88%E6%B0%B4%E5%8D%B0%E7%89%88%EF%BC%89.pdf
> > > > >
> > > > > [3] Towards Data Contamination Detection for Modern Large Language Models: Limitations, Inconsistencies, and Oracle Challenges, Samuel et al., 2024

---

### Official Review · Reviewer_eaft · 2025-10-30

**Soundness:** 2
**Presentation:** 3
**Contribution:** 2
**Rating:** 4
**Confidence:** 4

**Summary:**

This paper introduces OmniEduBench, a comprehensive Chinese benchmark designed to evaluate large language models (LLMs) in educational settings across two core dimensions: knowledge and cultivation. It comprises 24,602 question–answer pairs spanning 61 subjects (41 in knowledge, 20 in cultivation) and 11 common exam question types. Experiments on 11 mainstream LLMs reveal significant performance gaps: even the best model, Gemini-2.5 Pro, achieves only 62.76% accuracy in the knowledge dimension, while the top performer in cultivation (QwQ) lags nearly 30% behind human-level performance. The results underscore the challenges of deploying LLMs as effective educational assistants in real-world Chinese contexts.

**Strengths:**

1. The dataset is relatively large in scale.

2. The experimental setup is rigorous: both 0-shot and few-shot evaluations are conducted separately, and the impact of using different LLMs as scoring models on the results is compared.

**Weaknesses:**

1. There is insufficient detailed comparison with existing educational benchmark datasets (e.g., C-Eval) in terms of data volume, dimensions, etc.

2. The description of the “HARD” subset is limited. If it is defined as “the 26% of samples on which models perform worst,” this may bias the subset toward the weaknesses of specific models rather than reflecting objective difficulty.

3. The paper should include more details about model inference settings—such as temperature, sampling strategy, and whether results are averaged over multiple runs—to better demonstrate the fairness of the evaluation.

4. The baselines lack specialized expert models in specific domains, such as those tailored for mathematics or medical tasks.

**Questions:**

1. How is the OmniEduBench HARD subset constructed and annotated? Is there a clear description of whether difficulty is determined based on human expert judgment or model performance?

2. As shown in Tables 3 and 4, smaller Qwen3 models outperform larger ones. Has the authors conducted any specific analysis to explain why model performance on OmniEduBench deviates from the expected scaling law?

3. In the Chinese context, several models claim to support educational applications (e.g., ERNIE and iFlytek Spark). Why were these models not included in the evaluation?

4. Cultivation dimension is an abstract concept—how does this paper ensure the reasonableness and representativeness of the tasks designed for this dimension?

---

> ### Author Response · Authors · 2025-11-21
> **Responses by Authors (Part 1)**
>
> > **[W1] There is insufficient detailed comparison with existing educational benchmark datasets (e.g., C-Eval) in terms of data volume, dimensions, etc.**
>
> **Response [W1]:** We thank the reviewer for pointing out this detail. To more clearly illustrate the differences between existing datasets and our OmniEduBench, **we conducted a systematic comparison across seven dimensions: data dimension, data size, data type, data source, data domain, covered subjects, and data language.** The comparison results have been organized and updated in Table 2 of the paper. The comparison results show that OmniEduBench exhibits clear advantages over existing benchmarks across all evaluated dimensions. Here, Mixed (X) indicates X types of mixed question types, and General (K) signifies that the dataset covers all subjects and educational stages with K categories. ZH and EN represent Chinese and English, respectively.
>
> | Benchmark | Dimension | Size | Type | Data source | Domain | Subject | Language |
> | :--- | :--- | :--- | :--- | :--- | :--- | :--- | :--- |
> | C-Eval [1] | Knowledge | 13.9K | Multiple answer | Publicly available | General (4) | 52 | ZH |
> | EduBench [2] | Knowledge | 18.8K | Multiple choice | LLM-generated | General (9) | 9 | ZH & EN |
> | CMATH [3] | Knowledge | 1.7K | Multiple choice | Publicly available | Math | 1 | ZH |
> | E-EVAL [4] | Knowledge | 4.35K | Multiple answer | Publicly available | K-12 | 23 | ZH |
> | GAOKAO [5] | Knowledge | 2.8K | Mixed (7) | Publicly available | High school | 9 | ZH |
> | GSM8K [6] | Knowledge | 8.5K | Multiple choice | Publicly available | Math | 1 | EN |
> | MATH [7] | Knowledge | 12.5K | Multiple choice | Publicly available | Math | 7 | EN |
> | OmniEduBench (ours) | Knowledge & Cultivation | 24.6K | Mixed (11) | Publicly & Private & LLM | General (12) | 61 | ZH |
>
> > **[W2] The description of the “HARD” subset is limited. If it is defined as “the 26% of samples on which models perform worst,” this may bias the subset toward the weaknesses of specific models rather than reflecting objective difficulty.**
>
> **Response [W2]:**  We provide further clarification of the original description here. The “HARD” subset in OmniEduBench refers to the 26% of samples on which all 11 mainstream open-source and closed-source models we evaluated perform relatively poorly, rather than reflecting the weaknesses of any specific model. In addition, we invited 10 master’s students to manually review the selected HARD samples to verify whether these problems indeed possess higher inherent difficulty. The review results show that the samples on which models tend to fail are largely consistent with human-perceived difficulty. Therefore, the HARD subset can objectively reflect the actual difficulty of the problems rather than being a consequence of model bias. We have also updated the corresponding description at lines 428-463 in the paper.
>
> > **[W3] The paper should include more details about model inference settings—such as temperature, sampling strategy, and whether results are averaged over multiple runs—to better demonstrate the fairness of the evaluation.**
>
> **Response [W3]:** Following C-Eval [1] and GSM8K [6], we set the temperature to T=0 during model inference, which corresponds to using greedy decoding. At each generation step, the model selects the token with the highest probability, ensuring deterministic outputs that minimize randomness and produce the most common and stable answers. Nonetheless, we conducted three independent runs and averaged the results. Due to T=0, the results across the three runs showed negligible differences. Details of the experimental setup have been updated in lines 374-377 of the paper.
>
> > **[W4] The baselines lack specialized expert models in specific domains, such as those tailored for mathematics or medical tasks.**
>
> **Response [W4]:** (1) Our baseline settings are based on C-eval [1] and EduBench [2], with additional comparisons including more recent LLMs and the recently open-sourced education model MuduoLLM [3].
>
> (2) OmniEduBench covers a comprehensive range of subjects from primary school, middle school, university to vocational education. Therefore, comparing baselines on a specific subject such as mathematics or medicine may be somewhat unfair. To further address the reviewer’s concerns, we have also provided relevant experimental results as follows and updated in Tables 4 and 5. Experimental results show that domain-specific baselines (*e.g.*, those designed for subjects such as mathematics or medicine) perform poorly when evaluated on a comprehensive, multi-disciplinary dataset.

---

> ### Author Response · Authors · 2025-11-21
> **Responses by Authors (Part 2)**
>
> | Knowledge dimension | FD | HH | SSEM | LP | MH | IIS | Average |
> | :--- | :--- | :--- | :--- | :--- | :--- | :--- | :--- |
> | ERNIE-4.5-Turbo  | 39.89 | 51.59 | 45.61 | 52.07 | 59.15 | 47.43 | 49.29  |
> | iFlytek Spark_Pro | 22.96 | 37.03 | 41.58 | 41.03 | 36.82 |  37.90 | 36.22 |
> | HuatuoGPT-o1-7B (Medicine)  | 23.67 | 24.23 | 23.38 | 18.79 | 29.74 | 20.94 |23.46 |
> | Qwen2.5-Math-7B-instruct (Math) | 33.07 | 8.43 | 11.51 | 8.39 | 10.67 | 13.93 | 14.33 |
>
> | Cultivation | TCS | EMH | SIS | CV | PD | TFS | Average |
> | :--- | :--- | :--- | :--- | :--- | :--- | :--- | :--- |
> | ERNIE-4.5-Turbo |70.11 | 62.85 | 66.30| 56.77| 67.02 | 70.47| 65.65|
> | iFlytek Spark_Pro | 68.05| 61.87| 64.95| 61.10| 67.80 | 72.02 | 65.24|
> | HuatuoGPT-o1-7B (Medicine) | 42.58| 45.57| 48.10|41.79|43.65|52.85|44.44|
> | Qwen2.5-Math-7B-instruct (Math) |28.74|31.44|20.52|27.52|16.97|22.28|26.34|
>
> > **[Q1] How is the OmniEduBench HARD subset constructed and annotated? Is there a clear description of whether difficulty is determined based on human expert judgment or model performance?**
>
> **Response [Q1]:** The "HARD" subset in OmniEduBench refers to the 26% of samples on which all 11 mainstream open-source and closed-source models we evaluated performed poorly, rather than reflecting the weaknesses of any specific model. Additionally, we invited 10 master’s students to manually review the selected HARD samples to verify whether these questions indeed possess higher inherent difficulty. The review results showed that the samples where models tended to err were largely consistent with human-perceived difficulty. In summary, the difficulty is determined based on both model performance and human expert judgment. Therefore, the HARD subset can objectively reflect the actual difficulty of the questions rather than being a result of model bias. We have updated the corresponding description at lines 428-463 in the paper.
>
> > **[Q2] As shown in Tables 3 and 4, smaller Qwen3 models outperform larger ones. Has the authors conducted any specific analysis to explain why model performance on OmniEduBench deviates from the expected scaling law?**
>
> **Response [Q2]:** We appreciate the reviewer’s attention. Regarding the phenomenon that smaller Qwen3 models outperform larger ones on OmniEduBench, we have conducted preliminary analysis and believe the possible reasons include:
> (1) Differences in overfitting and generalization: Although larger models have more parameters, they may overfit to the specific tasks and data distribution of this benchmark, resulting in reduced generalization ability.
> (2) Training details and optimization strategies: Models of different sizes may have differences in training procedures, hyperparameter tuning, and optimization strategies, which affect their actual performance.
> (3) Data and task suitability: The types and difficulty levels of questions covered by OmniEduBench may better match the capabilities of certain model sizes, so increasing model size does not necessarily lead to linear performance gains.
> **We plan to further investigate this phenomenon in future work, including broader comparisons across model sizes and architectures as well as more detailed performance analyses.**
>
> > **[Q3] In the Chinese context, several models claim to support educational applications (e.g., ERNIE and iFlytek Spark). Why were these models not included in the evaluation?**
>
> **Response [Q3]:** Thank you for the reviewer's suggestion. We have included comparisons of these two models, as well as other models (see [W4]). Experimental results have been updated in Tables 4 and 5 of the paper. Even when comparing education-specific models, this performance highlights the challenging nature and strong discriminative power of the constructed OmniEduBench.
>
> | Knowledge | FD | HH | SSEM | LP | MH | IIS | Average |
> | :--- | :--- | :--- | :--- | :--- | :--- | :--- | :--- |
> | ERNIE-4.5-Turbo_ans | 39.89 | 51.59 | 45.61 | 52.07 | 59.15 | 47.43 | 49.29|
> | iFlytek Spark_Pro_ans | 22.96 | 37.03| 41.58 | 41.03 | 36.82 |  37.90 | 36.22|
>
> | Cultivation | TCS | EMH | SIS | CV | PD | TFS | Average |
> | :--- | :--- | :--- | :--- | :--- | :--- | :--- | :--- |
> | ERNIE-4.5-Turbo |70.11 | 62.85 | 66.30| 56.77| 67.02 | 70.47| 65.65|
> | iFlytek Spark_Pro | 68.05| 61.87| 64.95| 61.10| 67.80 | 72.02 | 65.24|

---

> ### Author Response · Authors · 2025-11-21
> **Responses by Authors (Part 3)**
>
> > **[Q4] Cultivation dimension is an abstract concept—how does this paper ensure the reasonableness and representativeness of the tasks designed for this dimension?**
>
> **Response [Q4]:** We fully agree with the reviewer’s opinion. Regarding the cultivation dimension, we have planned content covering 20 subjects, primarily following these principles: (1) **Design with multiple experts**: we invited experts and scholars in the field of education, especially those focusing on cultivation development, to participate jointly. (2) **Solid theoretical foundation**: the design of the cultivation dimension is based on cultivation development guidelines issued by schools or regions, combined with relevant curricula and standards. (3) **Close alignment with actual teaching**: subject selection is closely integrated with real teaching content, textbooks, and typical classroom scenarios to guarantee the subjects’ practical significance and representativeness, accurately reflecting the key abilities students need to master during the learning process.
>
> [1] C-Eval: A multi-level multi-discipline Chinese evaluation suite for foundation models, NeurIPS, 2023.
>
> [2] EduBench: A Comprehensive Benchmarking Dataset for Evaluating Large Language Models in Diverse Educational Scenarios, arXiv 2025.
>
> [3] CMATH: Can Your Language Model Pass Chinese Elementary School Math Test?, arXiv 2023.
>
> [4] E-Eval: A comprehensive Chinese K-12 education evaluation benchmark for large language models, arXiv preprint arXiv 2024.
>
> [5] GAOKAO: Evaluating the performance of large language models on the gaokao benchmark, arXiv 2023.
>
> [6] GSM8K: Training Verifiers to Solve Math Word Problems, arXiv 2021.
>
> [7] MATH: Measuring mathematical problem solving with the math dataset, arXiv 2021.
>
> [8] Beijing Normal University and TAL Education Group, https://smartedu-bnu.tal.com/, accessed on July 14, 2025.

---

### Official Review · Reviewer_bj4W · 2025-11-02

**Soundness:** 3
**Presentation:** 3
**Contribution:** 2
**Rating:** 4
**Confidence:** 4

**Summary:**

This paper introduces OmniEduBench, a Chinese education benchmark designed to evaluate LLMs on both knowledge understanding and cultivation skills. OmniEduBench consists of 24.602K question-answer pairs with 18.121K for knowledge and 6.481K entries for cultivation. This benchmark features a 11 common exam question types and has done extensive experiments on 11 mainstream open-source and closed-source LLMs. The benchmark highlights room for improvement of applying LLMs in education.

**Strengths:**

1.	The paper introduces a highly comprehensive benchmark, commendable for its broad coverage spanning 41 distinct subjects and 11 common question types.
2.	OmniEduBench includes the "cultivation dimension," which moves beyond mere knowledge assessment to evaluate crucial pedagogical competencies, including emotional support and value guidance.
3.	The manuscript is well-organized and clearly written, making the methodology easy to follow.

**Weaknesses:**

1.	Serious Data Quality Concerns: The quality of the dataset is questionable, based on the paper's own examples. Several provided questions are flawed:
a)	Figure 3(b) (cultivation dimension) is highly subjective; option C appears to be an equally valid approach, making the single "correct" answer debatable.
b)	Figure 4(b) (knowledge dimension) presents a proof question in part (2) but provides no proof or expected solution.
c)	Figure 8 (value alignment) contains ambiguous "correct" options. It's unclear why options A (Q1) and D (Q2) are not considered valid, raising concerns about the benchmark's objectivity.
These examples damage confidence in the overall quality of the 24.6K items.
2.	The "dual-model filtering" approach to defining difficulty raises concerns about the benchmark's generalizability. By selecting only questions that two specific models(Qwen3 and QWQ) failed, the benchmark (OmniEduBench) risks being tailored to their particular weaknesses. As a result, it may not serve as a balanced test of general educational knowledge for other models.
3.	The dataset's heavy reliance on LLM-generated data (800k/927k) is a major concern. This approach risks using the LLM's own biases rather than objective, "correct values." The human annotator appears insufficient; it was limited to filtering by 50 students and scoring by 5 experts, with a lack of human-led question modification. Furthermore, the evaluation criteria for this human annotation are vague and not clearly explained in the paper.

**Questions:**

1.	The authors must clarify the specific data quality issues raised in Weakness 1. This clarification is crucial to establish the dataset's objectivity and resolve the existing doubts about its quality..
2.	The paper states 800K data points were LLM-generated. To assess reproducibility and potential bias, please provide more details. Which specific LLM(s) were used? What prompts or generation strategies were employed, particularly to create the nuanced scenarios for the "cultivation" dimension？
3.	The paper needs more detail on the human annotation process. What specific guidelines and criteria were given to the 50 students for the filtering task? Similarly, what were the detailed scoring rules used by the 5 senior annotators for the quality evaluation in Table 2.
4.	The paper mentions "LLM-assisted scoring." Please elaborate on this method. How exactly is the LLM used to evaluate responses (e.g., what prompts are used)? Crucially, how was the reliability and accuracy of this automated scoring validated, for instance, by correlating it against human expert scores?

---

> ### Author Response · Authors · 2025-11-21
> **Responses by Authors (Part 1)**
>
> > **[W1] Serious Data Quality Concerns: The quality of the dataset is questionable, based on the paper's own examples.**
>
> **Response [W1]:** We sincerely apologize for any misunderstanding the reviewer may have had regarding the quality of our dataset. We hope that by providing detailed responses to three specific concerns, we can alleviate the reviewer's worries about the dataset's quality.
>
> > **[W1] (a) Figure 3(b) (cultivation dimension) is highly subjective; option C appears to be an equally valid approach, making the single "correct" answer debatable.**
>
> **Response [W1] (a):** In designing the cultivation dimension, we intentionally craft distractors from superficially plausible options to rigorously assess the true cultivation teaching ability of LLMs. For instance, in Figure 3(b), Option C acts as a distractor to the correct Option D. Option D embodies the core cultivation principle that profound understanding stems from **personal experience and reflective engagement**, rather than mere instructional correction. Specifically, Option C, which involves a one-time teacher-student conversation to address mistakes, offers a valuable but potentially short-lived impact. Conversely, Option D advocates for students to **personally research and narrate the story of martyrs at a memorial site**. This active, firsthand experience is far more impactful than passive lecturing; it allows students to internalize the deeds and significance, thereby fostering enduring values and behaviors. In essence, while Option C represents sound pedagogical practice, Option D more deeply exemplifies cultivation education by transforming errors into opportunities for profound personal reflection, cultivating responsibility, and internalizing core values through direct engagement.
>
> > **[W1] (b) Figure 4(b) (knowledge dimension) presents a proof question in part (2) but provides no proof or expected solution.**
>
> **Response [W1] (b):** Thank you for pointing this out. Please be assured that Figure 4(b) in the OmniEduBench dataset does include a detailed proof. For brevity in the main paper, we presented a condensed version. We have now included the complete proof process in Figure 9 of the appendix for full transparency. It is important to note that all non-objective questions within OmniEduBench are accompanied by detailed explanations or proof processes to ensure clarity and verifiability.
>
> > **[W1] (c) Figure 8 (value alignment) contains ambiguous "correct" options. It's unclear why options A (Q1) and D (Q2) are not considered valid, raising concerns about the benchmark's objectivity. These examples damage confidence in the overall quality of 24.6K items.**
>
> **Response [W1] (c):**  Question Q1 in Figure 8 is designed to distinguish between basic empathy, compassion, and problem-solving under the cultivation dimension. Option A exhibits foundational empathy by correctly framing the child's behavior as a form of "self-protection." However, its proposal is fundamentally limited; it offers an emotional appeal without a professional methodology, making it likely to be disregarded as mere "emotionalism" in a professional context. Option B, conversely, represents a far more sophisticated approach. It demonstrates a superior form of empathy by focusing on the long-term, societal consequences of punitive actions like expulsion ("deepening hatred towards society"). More critically, it translates this concern into a professional and scientific problem-solving process with clear, actionable steps: (1) engage psychological counselors, (2) perform a behavioral assessment, and (3) formulate a data-driven intervention plan. This methodical approach elevates the response from a simple expression of sympathy to a concrete, collaborative, and evidence-based strategy. By doing so, it not only addresses the child’s immediate needs but also models the highest standard of educational practice, making it the unequivocally superior answer.

---

> > ### Author Response · Authors · 2025-11-21
> > **Responses by Authors (Part 2)**
> >
> > Question Q2 in Figure 8 is designed to differentiate between surface-level interest-driven learning and deep, transformative knowledge transfer within the cultivation dimension. Option D exemplifies a common, effective strategy for sparking initial interest: linking popular songs to classical poetry. The core task, "comparing artistic conceptions," primarily engages students in comprehension and associative thinking. While valuable, this represents a lower level of knowledge transfer, where knowledge is analyzed but not necessarily internalized or applied creatively in new contexts. Option C, however, demonstrates a more profound educational approach. It moves beyond mere interest and fosters intrinsic motivation by assigning a practical, personal purpose to learning poetry—as a source of self-encouragement. This reframes the text from an object of study to a tool for life. It actively guides students toward the highest level of knowledge transfer: the application of poetic wisdom, emotions, and philosophies to their own lives, as seen in diaries, essays, and personal reflections. This process of internalization and practical application is the hallmark of meaningful learning. Therefore, Option C is the superior choice because it perfectly integrates purpose (application) and utility (addressing real-life needs), leading to a more enduring and transformative educational experience.
> >
> > > **[W2] The "dual-model filtering" approach to defining difficulty raises concerns about the benchmark's generalizability. By selecting only questions that two specific models(Qwen3 and QWQ) failed, the benchmark (OmniEduBench) risks being tailored to their particular weaknesses. As a result, it may not serve as a balanced test of general educational knowledge for other models.**
> >
> > **Response [W2]:** We sincerely thank the reviewer for the valuable comments. Below, we further clarify the methodology and motivation behind our dual-model filtering approach, and explain the analyses we conducted to ensure the generalizability of OmniEduBench.
> >
> > **(1) Clarification of the dual-model filtering methodology: our goal is to identify “universally difficult problems,” not “model-specific weaknesses.”** We selected QWQ-32B and Qwen3-235B as filtering models because we view these state-of-the-art (SOTA) LLMs as proxies for current model capabilities. Our assumption is that if an educational problem cannot be solved by both of these strong models, it likely reflects a general bottleneck in today’s AI—such as complex logical reasoning or deep causal analysis—rather than an artifact of any specific model architecture or training data. In other words, by **using SOTA models as a high-level “filter,” we efficiently extract from large-scale question pools those “hard-core problems” that challenge the entire current LLM paradigm**. This enables us to construct a “high ceiling” benchmark capable of measuring future progress.
> >
> > **(2) Evidence supporting the benchmark’s generalizability.**
> > To verify that the difficulty of the selected questions is universal—and not specific to the Qwen family—we conducted the following analyses:
> >
> > **(2-1) Cross-model performance consistency:** We evaluated 11 LLMs of different architectures and sizes on OmniEduBench. These models uniformly exhibited low performance. This indicates that OmniEduBench captures shared weaknesses across models, rather than model-specific quirks.
> >
> > **(2-2) Human expert evaluation of question difficulty:** Manual inspection showed that the selected problems predominantly involve high-level cognitive skills, such as multi-step logical deduction and complex calculations. These types of questions are objectively challenging for both models and humans. Educational experts also conducted sample evaluations and confirmed the high cognitive complexity and educational value of the questions.
> >
> > **(2-3) Comparison with single-model filtering:** Compared with single-model filtering, using two top-performing models from different architectural families as a “dual verification” mechanism more effectively eliminates incidental or model-specific failures. This helps retain only the essential and universally challenging problems, thereby enhancing the fairness and representativeness of the benchmark.
> >
> > **(3) Additional advantages of the method.**
> >
> > **(3-1) Ensuring question quality and preventing data contamination:** Both models possess strong world knowledge and fact-checking ability. If a question is clearly understood but cannot be answered correctly, this strongly suggests that it is well-posed, unambiguous, and unlikely to have appeared in their training data (otherwise the models would have memorized it).
> >
> > **(3-2) High efficiency and scalability:** Compared with manual annotation or handcrafted difficulty rules, this method is highly automated and scalable, enabling the rapid construction of large-scale, high-difficulty benchmarks.

---

> ### Author Response · Authors · 2025-11-21
> **Responses by Authors (Part 3)**
>
> > **[W3] The dataset's heavy reliance on LLM-generated data (800k/927k) is a major concern. This approach risks using the LLM's own biases rather than objective, "correct values." The human annotator appears insufficient; it was limited to filtering by 50 students and scoring by 5 experts, with a lack of human-led question modification. Furthermore, the evaluation criteria for this human annotation are vague and not clearly explained in the paper.**
>
> **Response [W3]:** We apologize for the confusion caused to the reviewers. Since the quality of LLM-generated data is difficult to fully guarantee, we initially generated a large amount of data (about 800K). However, after the complete data processing pipeline, **we obtained a final set of 24.602K high-quality samples, namely OmniEduBench. Among them, 26.1% were sourced from publicly available data, 48.1% from private data, and the remaining 25.8% from LLM-generated data**.
>
> As shown in Table 2, we first invited 50 master's students to rate the data based on five dimensions — Overall Quality, Clarity, Option Perplexity, Accuracy, and Cultivation Value — using a 5-point scale, and removed data with scores below 3 in any dimension and in the overall average. After this first round of quality checks, we invited 5 senior verification experts (such as teachers) to conduct a rigorous quality review on a 15\% random sample drawn separately from publicly available, private, and LLM-generated data of OmniEduBench. Data entries that received a score below 3 were subsequently removed, thereby further improving the overall quality and reliability of the dataset.
>
> > **[Q1] The authors must clarify the specific data quality issues raised in Weakness 1. This clarification is crucial to establish the dataset's objectivity and resolve the existing doubts about its quality.**
>
> **Response [Q1]:** We agree with the reviewer’s point. For more detailed responses, please see [W1].
>
> > **[Q2] The paper states 800K data points were LLM-generated. To assess reproducibility and potential bias, please provide more details. Which specific LLM(s) were used? What prompts or generation strategies were employed, particularly to create the nuanced scenarios for the "cultivation" dimension？**
>
> **Response [Q2]:** We answer these questions point by point: (1) **Which specific LLM was used?** For generating the 800K data points, we employed the Qwen3-235B-A22B-Thinking-2507 model. (2) **What generation strategies were used?** During data generation, we first collaborated with local schools to collect a batch of professional materials, including policies, books, papers, and teacher manuals, in the cultivation domain, totaling approximately 50GB. Together with cultivation experts, we divided this content into 20 subjects based on cultivation dimensions. The model first traverses the content corresponding to each subject within the 50GB, then the data is tokenized, ensuring it does not exceed the model’s maximum token limit. Subsequently, the model generates questions based on the cultivation knowledge points and cases contained in the materials. To ensure that the questions have a certain level of challenge, we instructed the model to include plausible but professionally unsupported distractor options when setting the answer choices.
>
> > **[Q3] The paper needs more detail on the human annotation process. What specific guidelines and criteria were given to the 50 students for the filtering task? Similarly, what were the detailed scoring rules used by the 5 senior annotators for the quality evaluation in Table 2.**
>
> **Response [Q3]:** We need to clearly explain to the reviewers that the data collected from the three platforms in OmniEduBench are complete question-answer pairs, so there is no need for manual annotation of the data, only expert verification of data quality. As shown in Table 2, we first invited 50 master's students to rate the data based on five dimensions —**Overall Quality, Clarity, Option Perplexity, Accuracy, and Cultivation Value** — using a **5-point scale**, and removed data with scores below 3 in any dimension and in the overall average. After this first round of quality checks, we invited 5 senior experts (such as teachers) to conduct a second, rigorous review of sampled data, scoring according to the same dimensions, and further removing data scoring below 3, thereby further improving the overall quality of the dataset.

---

> > ### Author Response · Authors · 2025-11-21
> > **Responses by Authors (Part 4)**
> >
> > > **[Q4] The paper mentions "LLM-assisted scoring." Please elaborate on this method. How exactly is the LLM used to evaluate responses (e.g., what prompts are used)? Crucially, how was the reliability and accuracy of this automated scoring validated, for instance, by correlating it against human expert scores?**
> >
> > **Response [Q4]:** (1) **Elaborate the LLM-assisted scoring method:** The LLM-assisted scoring method refers to the approach commonly known as "LLM-as-judge" [1], where LLMs are employed as evaluators to assess responses in complex tasks. This method leverages the advanced reasoning and language understanding capabilities of LLMs to provide automated, scalable, and consistent evaluations, particularly useful for tasks that are difficult to assess using traditional metrics.
> >
> > (2) **About the LLM used to evaluate responses:**  The prompt used is as follows:  你是一个严格的判题助理。只输出JSON，不要解释。判断各模型作答是否与标准答案语义等价 (数值等价、单位同义等视为正确)。返回形如 {\"Evaluation\":{字段名: true/false}}，其中“字段名”必须与输入完全一致。（translated to English for clarity：You are a strict grading assistant. Only output JSON, no explanations. Judge whether each model’s answer is semantically equivalent to the standard answer (numerical equivalence and synonymous units are considered correct). Return a JSON object of the form {"Evaluation":{field_name: true/false}}, where 'field_name' must exactly match the input.）
> >
> > (3) **About the reliability and accuracy of this automated scoring:** "LLM-as-judge" or "LLM-assisted scoring" has become a commonly used evaluation method by researchers when dealing with complex tasks. To mitigate potential issues arising from relying on a single LLM as the evaluator, we present a comparison of scoring results from different LLMs as judges in Table 7 of the paper. Additionally, we invited 50 master’s students to manually review a 15% random sample to verify the accuracy and reliability of the LLM-assisted scoring.
> >
> > [1] A Survey on LLM-as-a-Judge. Arxiv 2025.

---

> > > ### Comment · Reviewer_bj4W · 2025-11-26
> > >
> > > Thanks for the response. I am still concerned about the benchmark's subjectivity (W1). It seems some good and practical answers are marked as incorrect simply because another option is considered more "ideal" . Given that , I decide to maintain the score and suggest changing these subjective single-answer questions to multiple-choice to improve the benchmark's validity. I look forward to further discussion of this submission.

---

> > > > ### Author Response · Authors · 2025-11-27
> > > > **Further responses the benchmark's subjectivity (Part 1)**
> > > >
> > > > > **[W1] I am still concerned about the benchmark's subjectivity (W1). It seems some good and practical answers are marked as incorrect simply because another option is considered more "ideal". Given that, I decide to maintain the score and suggest changing these subjective single-answer questions to multiple-choice to improve the benchmark's validity. I look forward to further discussion of this submission.**
> > > >
> > > > **Response [W1]:** We appreciate your valuable suggestion. We fully understand your concern that "some practical options are marked as incorrect," which indeed represents a significant challenge in subjective evaluation. However, regarding your proposal to convert questions into a multiple-choice (multi-select) format, after careful deliberation, we maintain that the "Single Best Answer" (single-choice) design is critical for precisely assessing model alignment with high-level educational goals in the cultivation dimension. We base this decision on the following rationales:
> > > >
> > > > **1. Value Prioritization via the "Single Best Answer" Principle:** In the assessment of the Cultivation Dimension, our core objective is not merely to verify how many options a model finds acceptable based on "common sense," but to examine its priority in value judgment. Distractors are often intentionally designed to represent "utilitarian," "short-term," or "partially correct" viewpoints, whereas the Ground Truth represents the highest value orientation aligned with national educational policies.
> > > >
> > > > The enforced single-choice format compels the model to perform implicit value ranking: it must discern which option strictly adheres to the high standards of "moral cultivation" among two seemingly reasonable choices. Switching to multi-select would allow models to select "practical but mediocre" options alongside the best one, thereby obscuring the model's ability to firmly select the "optimal solution" amidst critical value conflicts.
> > > >
> > > > **2. Ensuring Sensitivity and Discriminative Ability:** OmniEduBench is designed to be a high-standard benchmark. Adopting a multi-select format, where "imperfect but acceptable" options are counted as correct, effectively lowers the evaluation bar. This would result in inflated scores for models that are merely "basically reasonable" but lack "deep alignment," stripping the benchmark of its ability to distinguish top-tier models from average ones. While the single-choice format is rigorous, it effectively scrutinizes whether a model truly comprehends the nuances of core competencies and values defined within the educational system.
> > > >
> > > > **3. Expert-Validated "Uniqueness":** Although the concept of cultivation may appear abstract, it possesses clear standards under the policy framework we adhere to. As mentioned in our previous response, all items were rigorously constructed and validated by experts based on the educational syllabus. The expert team explicitly ruled out those "seemingly practical" distractors, confirming the uniqueness of the standard answer within the specific educational context. This design aims to establish a precise "Alignment Anchor," rather than facilitating an ambiguous, open-ended discussion.
> > > >
> > > > In conclusion, our adherence to the single-choice format is not for the sake of preserving exam form, but to ensure we can measure the degree of Fine-grained Alignment between models and high-standard educational values in the most sensitive manner possible.

---

### Comment · Area_Chair_iwec · 2025-11-24

Dear reviewers,

Thank you for your dedicated service as reviewers. Your efforts are critical to the success of our conference, and we deeply appreciate your time and expertise.

This paper has received reviews from reviewers but some have not provided a response to the author rebuttal. Given the limited time we have for author-reviewer discussions, we kindly ask you to share your post-rebuttal feedback to help clarify your perspective and aid the decision-making process.

Your input is invaluable in ensuring a fair and thorough review process.

Best,
AC

---

### Author Response · Authors · 2025-12-02
**General responses and manuscript revision summary**

Dear Area Chair and Reviewers,

We sincerely thank the Area Chair and all reviewers for their time and valuable feedback. We have carefully considered every comment and are pleased to report that we have thoroughly addressed all concerns in our rebuttal. We have revised the paper accordingly, with all changes highlighted in red. Below is a summary of our responses and the corresponding revisions:

* **Data quality**
  * ***Question:*** LLM-generated content may introduce model-induced biases. (Reviewers bj4w, Fmit, dsE8).
  * ***Response & Revision:*** OmniEduBench contains 24.602K high-quality samples, among which only 25.8% are LLM-generated. The remaining data are sourced from **reliable real-world education**. Moreover,  we further involved 55 human experts to perform independent verification, ensuring the quality in Table 3 and Section 3.2.

* **Definition and subjectivity of the cultivation**
  * ***Question:*** Reviewers expressed concerns that this metric is subjective and lacks definition. (Reviewers bj4w, eaft, Fmit)
  * ***Response & Revision:*** OmniEduBench is based on an in-depth study of authoritative national educational policy documents and provides a systematic interpretation of the cultivation. It concretizes the abstract cultivation requirements into assessable capabilities of models in core values, disciplinary literacy, and so on. We further collaborated with education experts to refine the cultivation into 6 major categories and 20 subjects (as shown in Figure 1). Meanwhile, we made an initial attempt to transform the broad cultivation into multiple-choice questions with standard answers. Through multiple rounds of expert cross-validation, we ensure that all correct options are rigorously verified (See Section 3.1).

* **OmniEduBench Hard**
  * ***Question:*** How is the OmniEduBench HARD subset constructed and annotated? (Reviewers bj4w, eaft)
  * ***Response & Revision:*** The *HARD* subset in OmniEduBench is defined as the 26% of samples on which all 11 mainstream open- and closed-source models we evaluated exhibited consistently poor performance. Thus, its construction does not reflect the weaknesses of any single model. We also invited 10 master’s students to conduct a manual review. Their assessments confirmed that the items frequently answered incorrectly by models largely aligned with human-perceived difficulty. This explanation has been added to Section 4.3.

* **More baselines**
  * ***Question:*** The reviewer suggested that we should include Chinese education-oriented baselines (such as ERNIE and Spark), as well as baselines from specific domains such as mathematics or medicine. (Reviewers eaft)
  * ***Response & Revision:*** Our baseline setup follows C-Eval and EduBench, and we further include recent LLMs and the newly open-sourced education model MuduoLLM. Since OmniEduBench spans subjects from primary, secondary, and higher education to vocational training, comparing baselines tailored to a single domain (*e.g.*, mathematics or medicine) would be inherently unfair. Nevertheless, to address the reviewer’s concern, we additionally report these domain-specific baselines in Tables 4 and 5. Results show that such specialized models perform poorly on a comprehensive, multi-disciplinary benchmark like OmniEduBench.

* **Compared with existing educational benchmarks**
  * ***Question:*** Reviewers pointed out that the paper lacks sufficiently detailed comparisons with existing educational benchmark datasets. (Reviewers eaft, Fmit)
  * ***Response & Revision:*** We conducted a systematic comparison across seven dimensions: data dimension, data size, data type, data source, data domain, covered subjects, and data language. The comparison results have been organized and updated in Table 2 of the paper. The comparison results show that OmniEduBench exhibits clear advantages over existing benchmarks across all evaluated dimensions.

* **Experimental details**
  * ***Question:*** The paper lacks sufficient details regarding model inference settings. (Reviewers eaft)
  * ***Response & Revision:*** Following C-Eval and GSM8K, we set the temperature to T=0, which corresponds to using greedy decoding. At each generation step, the model selects the token with the highest probability, ensuring deterministic outputs that minimize randomness and produce the most common and stable answers. Nonetheless, we conducted three independent runs and averaged the results. Details have been updated in Section 4.1.

* **Data contamination**
  * ***Question:*** The paper lacks verification of data contamination. (Reviewers Fmit)
  * ***Response & Revision:*** Nearly half of OmniEduBench (48%) comes from unpublished private corpora. We have also conducted data contamination checks following the reviewer. Both the verification results and the performance of 11 LLMs on OmniEduBench consistently demonstrate that our data quality is high and data contamination is minimal.

Thanks,

Submission 3730 Authors.

---

### Meta-Review · Area_Chair_D2h8 · 2026-01-02

**Summary:**

### Summary
This paper introduces OmniEduBench, a Chinese education benchmark designed to evaluate LLMs on both knowledge understanding and cultivation skills.

### Reviewer summary

Reviewers acknowledge the benchmark is large, diverse, and timely for Chinese educational applications, with relatively thorough model coverage.
The main concerns are dataset reliability, the subjectivity and single-best-answer design in the cultivation dimension, potential bias from LLM-generated content and LLM-as-judge scoring, and insufficiently convincing contamination checks and fairness of comparisons.

### AC Comments

The rebuttal adds clarifications and incremental evidence, including the final-source breakdown for the 24.6K set, the HARD subset definition, inference settings, additional baseline models, and contamination-related discussions.
However, I still tend to give Reject because the two most central risks remain:

1.  the cultivation dimension inherently embeds value judgments, and a reviewer explicitly maintains concerns that reasonable practical answers are marked incorrect under a “single best answer” policy; the rebuttal’s defense of single-choice does not resolve auditability and consensus issues.
2. for high-stakes education evaluation, the bar for answer correctness and labeling transparency is higher; the paper mostly provides process-level assurances and limited sampling checks rather than stronger, verifiable evidence.

Overall, this reads more like a scaled resource construction with improved documentation than a fully robust benchmark ready for adoption.

**Reviewer Concerns:**

### Reviewer bj4W

* Resolved: The authors explain why the example items have a unique best answer and clarify that subjective knowledge questions include detailed explanations or proofs, alongside more details on sources and human verification.
* Unresolved: The reviewer still flags cultivation subjectivity and suggests multi-select to improve validity; the authors insist single-choice is necessary for value ranking, but this does not address the core reproducibility and consensus issue.

### Reviewer eaft
* Resolved: added a systematic comparison table vs prior benchmarks, clarified HARD construction and inference settings, and added more baselines including some Chinese education-oriented models.
* Unresolved: The HARD subset is still model-performance filtered and may reflect shared model weaknesses rather than educational difficulty; broader coverage does not fix the core measurement validity concerns in cultivation.


### Reviewer Fmit
* Resolved: clarified the final 24.6K composition and added discussion on judge-model sensitivity and partial manual checks.
* Unresolved: The reviewer explicitly maintains their rating, citing answer correctness as essential, cultivation being subjective/under-defined, and contamination checks needing stronger black-box methodologies beyond source-isolation arguments.

### Reviewer dsE8
* Resolved: The authors clarified numerical claims, provided source breakdown, and added more fine-grained results by educational stage; the reviewer indicates their view remains positive.
* Unresolved: The broader benchmark-validity concerns remain, though this reviewer appears satisfied after clarifications.

**Reviewer Scores:**

dsE8, bj4W, Fmit participated the discussion, three of them stay the original rating (6,4,4).

if eaft participated the discussion, they would keep the negative score.

---

### Decision · Program_Chairs · 2026-01-26

Reject